# Structure and activation of the human autophagy-initiating ULK1C:PI3KC3-C1 supercomplex

Minghao Chen [1,2,3], Thanh N. Nguyen [3,4,5,6], Xuefeng Ren [1,2,3], Grace Khuu [3,4,5,6], Annan S. I. Cook [2,3,7], Yuanchang Zhao [1,8], Ahmet Yildiz [1,7,8], Michael Lazarou [3,4,5,6] & James H. Hurley [1,2,3,9] ✉

The Unc-51-like kinase protein kinase complex (ULK1C) is the most upstream and central player in the initiation of macroautophagy in mammals. Here, we determined the cryo-electron microscopy structure of the human ULK1C core at amino-acid-level resolution. We also determined a moderate-resolution structure of the ULK1C core in complex with another autophagy core complex, the class III phosphatidylinositol 3-OH kinase complex I (PI3KC3-C1). We show that the two complexes coassemble through extensive contacts between the FIP200 scaffold subunit of ULK1C and the VPS15, ATG14 and BECN1 subunits of PI3KC3-C1. The FIP200:ATG13:ULK1 core of ULK1C undergoes a rearrangement from 2:1:1 to 2:2:2 stoichiometry in the presence of PI3KC3-C1. This suggests a structural mechanism for the initiation of autophagy through formation of a ULK1C:PI3KC3-C1 supercomplex and dimerization of ULK1 on the FIP200 scaffold.

Macroautophagy (hereafter, 'autophagy') is the main cellular mechanism for the disposal of molecular aggregates and damaged or unneeded organelles[1]. While autophagy was first characterized as a bulk response to starvation, it is now clear that many forms of bulk and selective autophagy are central in development and cellular homeostasis[1]. The homeostatic necessity of autophagy is clearest in neurons, which are postmitotic and, thus, uniquely susceptible to toxic aggregates and damaged organelles[2]. Autophagic dysfunction has been linked to all major neurodegenerative diseases[2], with the clearest genetic linkage to Parkinson's disease[3].

All forms of canonical autophagy, bulk and selective, are initiated upon the recruitment and activation of the FIP200 protein[4] and the class III phosphatidylinositol 3-OH kinase complex I (PI3KC3-C1)[5–7].

FIP200 serves as the central scaffolding subunit of the Unc-51-like kinase protein kinase complex (ULK1C), providing a platform for the other three subunits: the ULK1 kinase, ATG13 and ATG101 (refs. 8–12). PI3KC3-C1 contains one copy each of the lipid kinase VPS34, the pseudokinase VPS15 and the regulatory subunits BECN1 and ATG14 (refs. 5,13). The former three subunits are also present in PI3KC3-C2 involved in endosomal sorting and late steps in autophagy, while ATG14 is uniquely involved in autophagy initiation[5–7]. Atomic-resolution and near-atomic-resolution structures are known for various fragments of these complexes[14–16] and structures with low-to-moderate resolution are known for human PI3KC3-C1 (refs. 17,18) and the core of human ULK1C (ref. 19).

The means by which ULK1C and PI3KC3-C1 activities are switched on and off are critically important for the physiology of autophagy initiation

[1]Department of Molecular and Cell Biology, University of California, Berkeley, Berkeley, CA, USA. [2]California Institute for Quantitative Biosciences, University of California, Berkeley, Berkeley, CA, USA. [3]Aligning Science Across Parkinson's (ASAP) Collaborative Research Network, Chevy Chase, MD, USA. [4]Walter and Eliza Hall Institute of Medical Research, Parkville, Victoria, Australia. [5]Department of Medical Biology, University of Melbourne, Melbourne, Victoria, Australia. [6]Department of Biochemistry and Molecular Biology, Biomedicine Discovery Institute, Monash University, Melbourne, Victoria, Australia. [7]Graduate Group in Biophysics, University of California, Berkeley, Berkeley, CA, USA. [8]Physics Department, University of California, Berkeley, Berkeley, CA, USA. [9]Helen Wills Neuroscience Institute, University of California, Berkeley, Berkeley, CA, USA. ✉e-mail: jimhurley@berkeley.edu

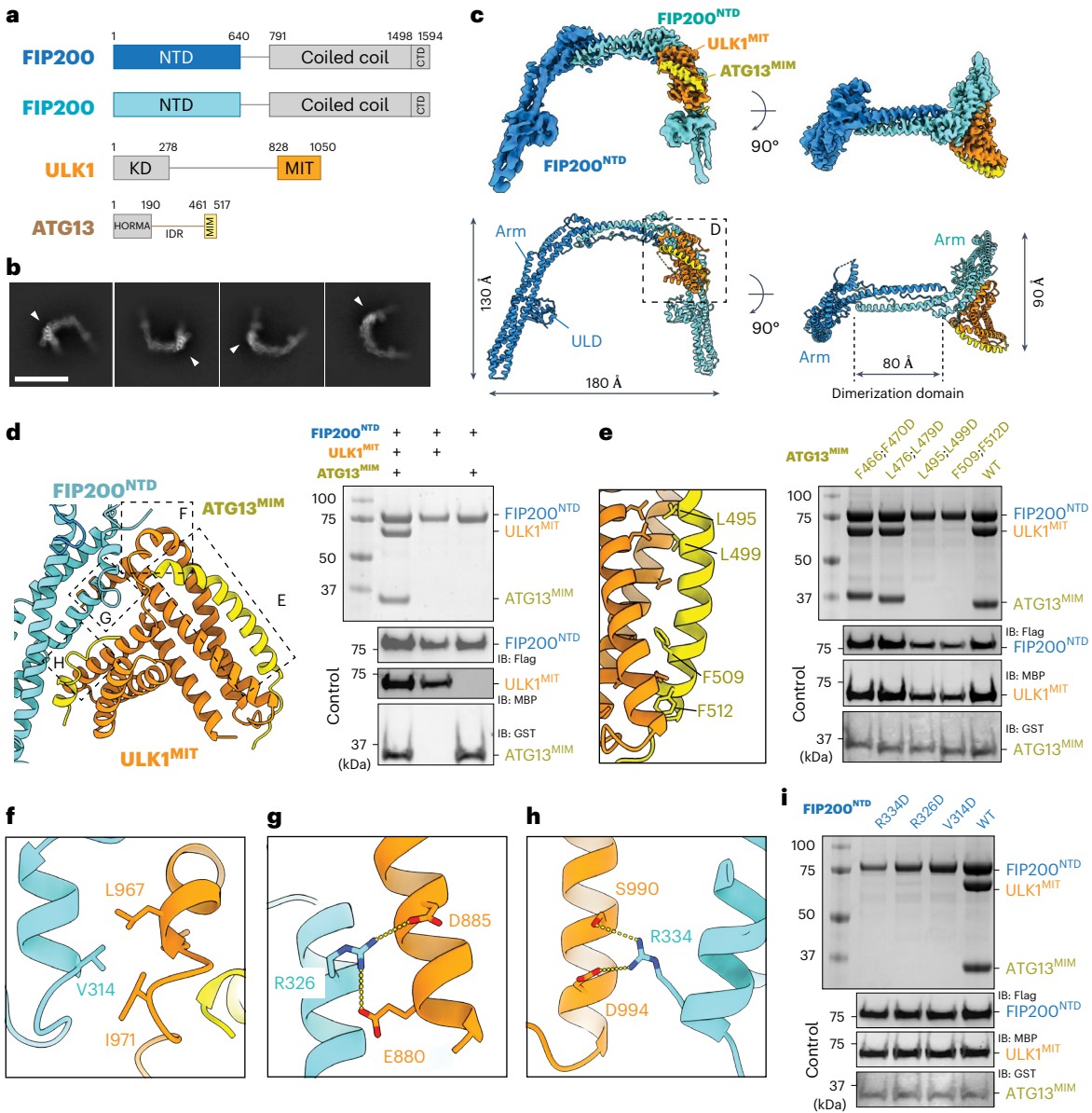

**Fig. 1 | Structure of the ULK1C core. a**, Domain organization of the ULK1C core. Gray indicates the regions truncated in this study. CTD, C-terminal domain; KD, kinase domain. **b**, Representative reference-free 2D class averages of the ULK1C core. Scale bar, 20 nm. **c**, Overview of the EM map (top) and the coordinates (bottom) of the ULK1C core. Subunits are depicted in the same color as in **a**. The map is contoured at 12σ. **d**, Pulldown assay of FIP200^NTD with ULK1^MIT and ATG13^MIM. Strep-Tactin resin was loaded with TSF-tagged FIP200^NTD to pull down MBP-tagged ULK1^MIT and GST-tagged ATG13^MIM in various combinations as indicated above the lanes. The experiment was repeated three times with similar results. **e**, Close-up view of the interface between ULK1^MIT and ATG13^MIM and the

corresponding Strep pulldown assay. FIP200^NTD–TSF was used to pull down various GST-tagged ATG13 mutants (F466;F470D, L476D;L479D, L495D;L499D and F509D;F512D) and MBP-tagged WT ULK1^MIT. The experiment was repeated four times with similar results. **f–h**, Close-up views of the interface between ULK1^MIT and FIP200. Key residues for the binding interface are indicated, with hydrogen bonds shown as yellow dotted lines. **i**, Strep pulldown assay for the FIP200–ULK1 interface. Various FIP200^NTD–TSF mutants (R334D, R326D and V314D) were used to pull down WT ULK1^MIT and ATG13^MIM. All pulldown results were visualized by SDS–PAGE and Coomassie blue staining. The experiment was repeated four times with similar results.

and for therapeutic interventions targeting neurodegeneration[20]. Yet these mechanisms have thus far been hidden because of the limitations of the available fragmentary or low-resolution structures. While ULK1C and PI3KC3-C1 are both critical for autophagy initiation and recruited at the earliest stages, beyond the presence of ULK1 phosphorylation sites on PI3KC3-C1 subunits[21–24], it has been unclear how their activities are coordinated. Here, we report the structures of the human ULK1C core at resolutions adequate for amino-acid-level interpretation. We also show that ULK1C and PI3KC3-C1 form a physical supercomplex, determine its structure and show that ULK1C can undergo a stoichiometric switch leading to the recruitment of two copies of the ULK1 kinase itself.

## Results

### Cryo-electron microscopy (cryo-EM) structure of a 2:1:1 stoichiometric ULK1C core

The ordered core of ULK1C was purified in a form consisting of the FIP200 N-terminal domain (NTD; 1–640) and the ULK1 C-terminal microtubule-interacting and transport (MIT) domain (836–1059) fused with the ATG13 residues (363–517) responsible for binding to FIP200 and ULK1 (ref. 19) (Fig. 1a). The peripheral domains, including the coiled-coil (CC; 791–1498) and claw (1499–1594) domains of FIP200, the kinase domain of ULK1 (1–278) and the ATG13 (1–190):ATG101 (1–218) HORMA (Hop1p/Rev7p/MAD2) domain, were removed from

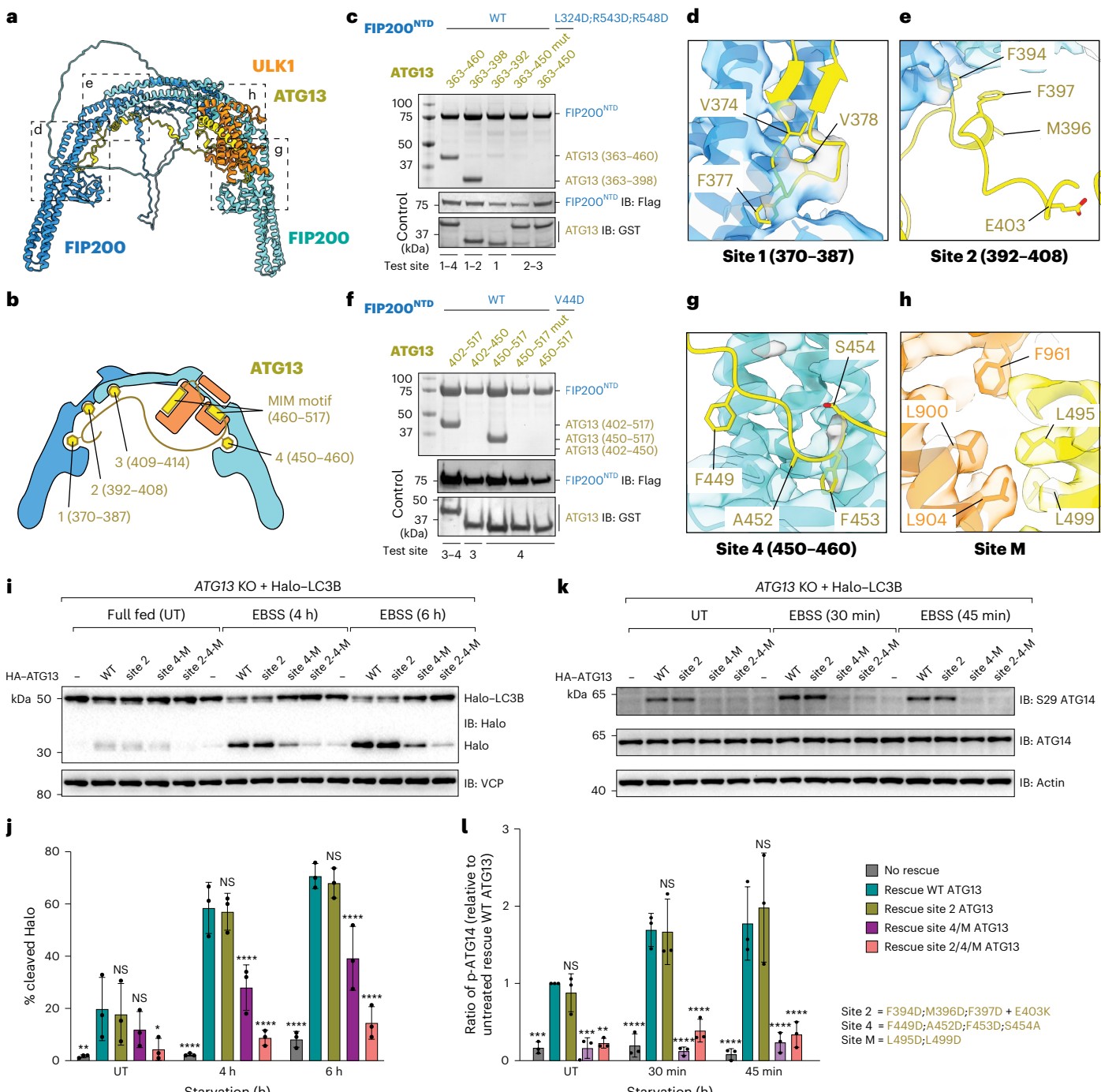

**Fig. 2 | Binding sites of ATG13^IDR on FIP200^NTD. a**, AlphaFold2 prediction model of the ULK1C core. **b**, Schematic diagram of the ULK1C core. **c**, Truncation and mutagenesis screening conducted by pulldown assay. Strep-Tactin resin was loaded with WT FIP200^NTD–TSF to pull down various ATG13 fragments (WT 363–460, 363–398, 363–392 and 363–450 with F394D;F397D;E403K substitutions (363–450 M1)) or loaded with FIP200^NTD–TSF variant (L324D;R543D;R548D) to pull down WT ATG13 fragment 363–460. The experiment was repeated four times with similar results. **d,e**, Close-up views of binding sites 1 and 2 of the AlphaFold2 model with the key binding residues shown. The map is contoured at 11σ. **f**, Strep pulldown assay. The WT FIP200^NTD–TSF was used to pull down various ATG13 fragments (WT 402–517, 402–450, 450–517 and 450–517 with A452D;F453D substitutions (450–517 M2)) or FIP200^NTD–TSF variant (V44D) was used to pull down WT ATG13 fragment 450–517. The results from the pulldown assays (**c,f**) were visualized by SDS–PAGE and Coomassie blue staining. The experiment was repeated four times with similar results. **g**, Close-up view of binding site 4 of the AlphaFold2 model. **h**, Close-up view of the ULK1^MIT-binding site (site M)

of ATG13^MIM determined by cryo-EM. The key binding residues are shown. **i,j**, *ATG13*-KO cells stably expressing Halo–LC3B without rescue or rescued with different versions of HA–ATG13 were treated with 50 nM TMR-conjugated Halo ligand for 15 min. Following that, cells were washed with 1× PBS and treated with EBSS for the indicated time periods, harvested and analyzed by immunoblotting with indicated antibodies (**i**) and the percentage of the cleaved Halo band was quantified (**j**). Data in **j** are the mean ± s.d. from three independent experiments. NS, not significant. ****$P < 0.0001$ (two-way ANOVA). The substitution sites of each variant and their corresponding residues are shown in the insert. **k**, *ATG13*-KO cells expressing Halo–LC3B without rescue or rescued with indicated versions of HA–ATG13 were left untreated or treated with EBSS for 30 min or 45 min. Cells were harvested and analyzed by immunoblotting with the indicated antibodies. **l**, The ratio of phosphorylated ATG14 (p-ATG14) relative to untreated rescue WT ATG13 samples was quantified. Data in **l** are the mean ± s.d. from three independent experiments. ****$P < 0.0001$ (two-way ANOVA). The definition of mutants (bottom right) applies to **i–l**.

the construct because they are flexibly connected to the ULK1C core through disordered loops and not known to participate in the ordered core assembly. Deletion of these domains did not affect the stability of the ULK1C core and facilitated its structure determination. A monodisperse peak from size-exclusion chromatography (SEC) was collected and used for cryo-EM data collection. Image processing and three-dimensional (3D) reconstruction resulted in a cryo-EM density map with a local resolution of 3.35 Å in the best regions, including the FIP200:ATG13:ULK1 interface (Extended Data Fig. 1a–g). The distal tips of the FIP200[NTD] molecules are mobile and the density there is less defined. The ATG13:ULK1 unit resembles the yeast Atg13[MIM] (MIT-interacting motif):Atg1[MIT] complex[25]; thus, we adopted the MIT/MIM terminology for the human complex. The quality of the density allowed assignment of amino acid residues of the ULK1[MIT] domain and ATG13[MIM] (Extended Data Fig. 1h–l). The structure confirmed the previous observation of a 2:1:1 FIP200:ATG13:ULK1 complex[19], while defining how ULK1[MIT] and ATG13[MIM] bind to FIP200 in amino-acid-level detail.

The ULK1C core has dimensions of 180 × 130 × 90 Å and contains two molecules of FIP200[NTD] and one molecule each of ULK1 and ATG13 (Fig. 1b,c). FIP200[NTD] forms a C-shaped dimer as seen at low resolution[19], which can now be described in detail with sequence assigned. The two arms of FIP200 consist of 120-Å-long bundles formed by three twisted helixes (81–495). The two arms are connected by an 80-Å-long dimerization domain (496–599) bent at nearly 90° to the arms, resulting in the C shape. The structure after residue 599 was not resolved because of presumed disorder. The FIP200 dimer is in some ways reminiscent of the structure of its yeast counterpart, Atg17 (ref. [26]), although the protein folds are distinct. Atg17 also dimerizes through two arms that are in the same plane, although the arms are arranged in an S shape instead of a C (Extended Data Fig. 2a). Residues 1–80 of FIP200 form a ubiquitin-like domain (ULD) located at the middle of the arm domain on the inner side of the C shape. The higher-resolution analysis here confirms the previous observation that the ULD and the arm domain of FIP200 have the same fold as the scaffold domain of the TANK-binding kinase 1 (TBK1)[19] (Extended Data Fig. 2b), which is itself central to the initiation of some forms of autophagy[27].

Cryo-EM density for one copy of a ULK1[MIT]:ATG13[MIM] heterodimer was observed on one 'shoulder' of the FIP200 dimer. The ATG13[MIM] binds collaboratively to both FIP200 and ULK1[MIT] (Fig. 1d), consistent with the role of ATG13 in recruiting ULK1 (ref. [28]) downstream of FIP200. As seen for yeast Atg13:Atg1 (ref. [25]), the structure consists of two four-helix bundles (Extended Data Fig. 2c). Three helices of each are from the ULK1[MIT] and one from the ATG13[MIM]. Four pairs of hydrophobic residues (F466 + F470, L476 + L479, L495 + L499 and F509 + F512) from the ATG13[MIM] form prominent direct interactions with ULK1[MIT] (Extended Data Fig. 2d,e). Mutagenesis and pulldown assays confirmed that the latter two pairs, L495 + L499 and F509 + F512, are critical for the binding (Fig. 1e). The ULK1[MIT]:ATG13[MIM] heterodimer binds to FIP200 through two main interfaces (Fig. 1f–h). Three FIP200 residues, V314, R326 and R334, were identified by cryo-EM as potentially important for the ULK1:ATG13 interaction, which was confirmed by mutagenesis and pulldown assays (Fig. 1i).

### The ATG13[IDR]:FIP200[NTD] interface

The C-terminal portion of the ATG13 intrinsically disordered region (ATG13[IDR]), corresponding to residues 363–460, was previously demonstrated to bind to the FIP200[NTD] (ref. [19]) but was not visualized with sufficient continuity to assign amino acid sequence on the basis of the density alone. In this study, we used AlphaFold2 (ref. [29]), as corroborated by hydrogen–deuterium exchange mass spectrometry (HDX–MS)[19], to map four potential binding sites on ATG13 (site 1, 370–387; site 2, 392–408; site 3, 409–414; site 4, 450–460) (Fig. 2a,b and Extended Data Fig. 3). The sites augment the role of the ATG13[MIM] (460–517), referred to as site M hereafter. The interaction of a single ATG13[IDR] extends through both protomers of the FIP200[NTD] dimer.

Sites 1 and 4 are unique in ATG13 but their binding sites on two FIP200 protomers are structurally symmetric. One FIP200 molecule is occupied by ATG13 site 4 and the ULK1[MIT]:ATG13[MIM] heterodimer and the other FIP200 molecule is occupied by sites 1–3 of the ATG13[IDR] loop (Fig. 2b). This explains the unusual 2:1:1 stoichiometry of the ULK1C by showing in atomistic detail how the FIP200[NTD] dimer binds a single copy of ATG13 in the complex.

We screened a series of truncations to map the FIP200-binding determinants of ATG13[IDR]. Two ATG13 fragments, 363–398 (site 2) and 450–460 (site 4) were independently sufficient for FIP200 binding (Fig. 2c), consistent with the previous observation that these regions are protected from HDX–MS[19] (Extended Data Fig. 3b). Point mutagenesis showed that ATG13 residues F394, F397 and E403 and FIP200 residues L324, R543 and R548 are required for binding through site 2 (Fig. 2c–e), whereas ATG13 residues A452 and F453 and FIP200 V44 are key for binding through site 4 (Fig. 2f,g). We tested whether disrupting the binding between ATG13 and FIP200 would affect bulk autophagy in response to starvation. ATG13 variants (Fig. 2d,e,g,h) were introduced into *ATG13*-knockout (KO) HeLa cells (Extended Data Fig. 4). Mutations impacting site 2 (F394D;M396D;F397D;E403K) alone did not have a significant effect. Mutations impacting sites 4 and M (site 4, F449D;A452D;F453D;S454A; site M, L495D;L499D) led to a substantial reduction in flux, whereas mutations impacting sites 2, 4 and M together nearly eliminated flux (Fig. 2i,j). To investigate whether the reduction in autophagy is because of the decreased activity of ULK1C, we measured the phosphorylation level of ATG14[S29], a substrate of ULK1 kinase that has an important role in regulating the function of PI3KC3-C1 (ref. [23]). The results of the phosphorylation assay showed that reduced ULK1 activity correlates with reductions in autophagy flux. Mutations impacting site 2 alone had no significant effect, whereas combined mutations impacting sites 2, 4 and M eliminated ULK1 activity (Fig. 2k,l).

### ULK1C and PI3KC3-C1 form a supercomplex

PI3KC3-C1 is a substrate of ULK1 (refs. [21,23]). However, it has been unclear whether PI3KC3-C1 is recruited coordinately with ULK1C in autophagy initiation. To determine whether there was a direct interaction and, if so, with which portion of the complex, we assessed binding of PI3KC3-C1 to the ULK1C core and the ATG13–ATG101 HORMA dimer. Direct binding between FIP200 and PI3KC3-C1 was observed using a bead-binding assay (Fig. 3a,b). The ATG13–ATG101 HORMA dimer did not interact with PI3KC3-C1 under these conditions (Fig. 3a,b). ULK1C and PI3KC3-C1 were mixed and visualized by cryo-EM (Extended Data Fig. 5a). A small population of ULK1C core in complex with PI3KC3-C1 was observed in this dataset (Extended Data Fig. 5b). The mixture was then further purified by pulldown with PI3KC3-C1 (Fig. 3c) and subjected to cryo-EM (Fig. 4 and Extended Data Fig. 6). Two-dimensional (2D) classes of the ULK1C:PI3KC3-C1 supercomplex were visualized, in which the distinctive C-shaped density of the ULK1C core and the V-shaped density of PI3KC3-C1 were both clearly observed (Fig. 4a). Further 3D reconstruction resulted in an EM map containing the crescent FIP200 dimer and all four subunits (VPS34, VPS15, BECN1 and ATG14) of PI3KC3-C1 (Extended Data Fig. 6). The peripheral regions, including the FIP200 molecule binding with ULK1[MIT]:ATG13[MIM] and VPS34[HELCAT] (helical and catalytic domains), were not modeled because of their high mobility. In contrast, the regions containing VPS15[HSD] (helical solenoid domain), BECN1[CC]–ATG14[CC], BECN1[BH3] and the FIP200 arm not bound to ULK1[MIT]:ATG13[MIM] (hereafter, the 'proximal arm') were the best resolved parts, with local resolution at 6.84 Å (Extended Data Fig. 6h). This enabled unambiguous identification of the secondary structures identified in the atomistic structures of the PI3KC3-C1 core[30] and the proximal arm of FIP200 (Fig. 4b and Extended Data Fig. 7a).

Two main interfaces were identified between FIP200 and PI3KC3-C1 (Fig. 4c). The first interface is formed between the FIP200[(166–181, 473–485)] and BECN1[BH3] (Extended Data Fig. 7b), a well-studied

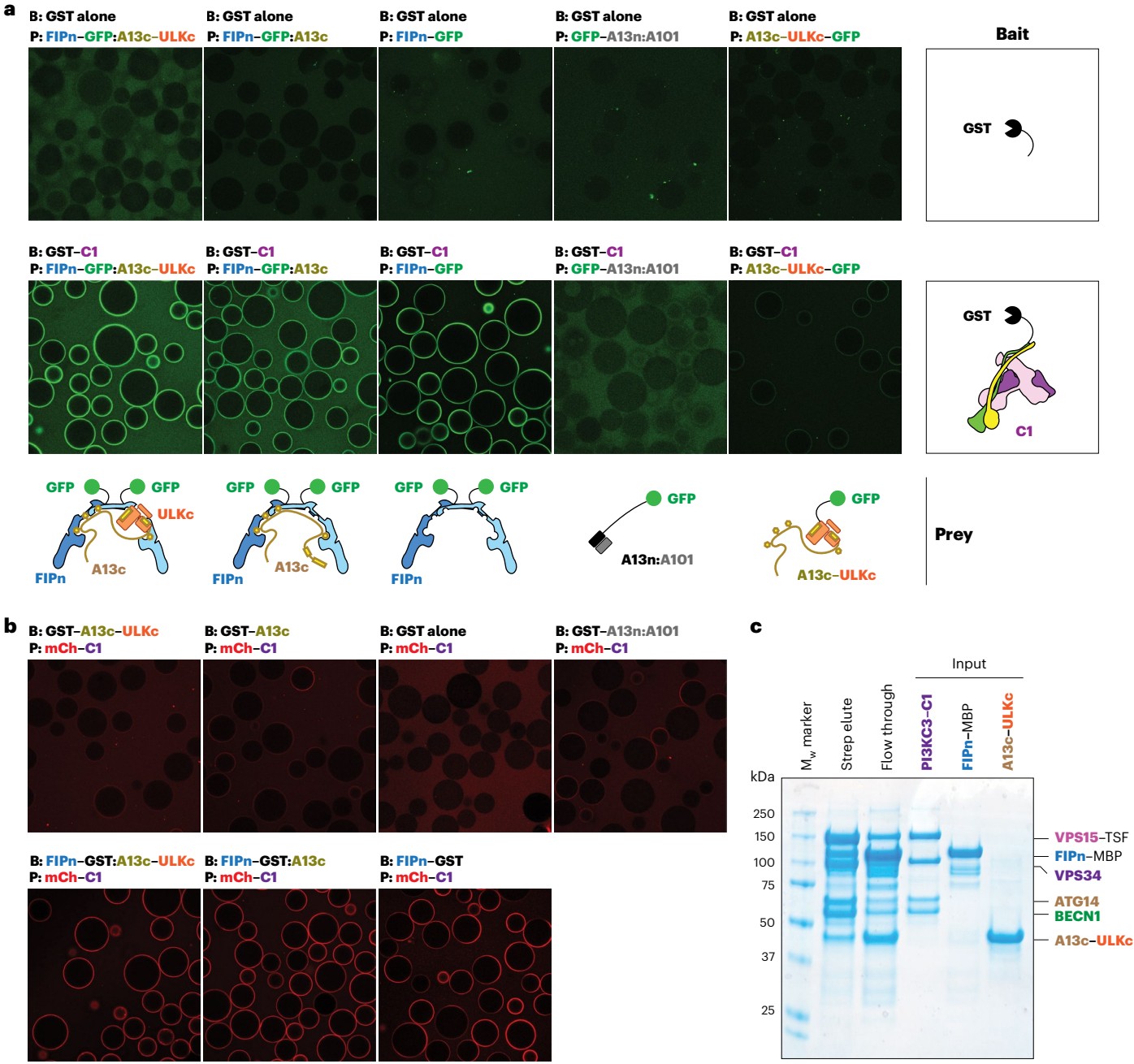

**Fig. 3 | Direct interaction between ULK1C and PI3KC3-C1. a,** Bead-binding assay of ULK1C core and PI3KC3-C1. GSH beads coated with GST tag alone or GST-tagged PI3KC3-C1 was used to recruit various combination of GFP-tagged proteins. The schematic drawing illustrates the bait and prey samples used for each condition. **b,** Complementary bead-binding assay for confirming the interaction between FIP200 and PI3KC3-C1. GST-tagged various proteins were used to recruit the mCherry-tagged PI3KC3-C1. **c,** SDS–PAGE of the ULK1C:PI3KC3-C1 pulldown sample. Strep resin was used to pull down the TSF-tagged VPS15. The eluate was obtained by washing the resin with buffer containing 10 mM D-desthiobiotin and visualized by SDS–PAGE and Coomassie blue staining. The experiment was repeated two times with similar results.

Bcl-2/Bcl-X$_L$-binding region that regulates autophagy and apoptosis[31–34]. A zinc-finger motif, composed of BECN1[C137, C140] and ATG14[C43, C46], is located behind the BH3 domain and supports the conformation. The BECN1–ATG14 zinc-finger and the BECN1 BH3 region were not visualized in previous PI3KC3-C1 structures; thus, it appears that the presence of the FIP200 proximal arm induces their ordering. The second interface is formed by the globular ULD domain of FIP200 and the curved HSD domain of VPS15 (Extended Data Fig. 7c). The VPS15[HSD] adjoins FIP200 sites 1 and 2, forming a large pocket with the FIP200 proximal arm and ULD. Fragmented EM density is observed in this pocket (Fig. 4d). Super-position of a predicted model of the ULK1C core onto the supercomplex

structure shows that the unassigned density could be contributed by ATG13[371–385]; however, the resolution is insufficient for unambiguous assignment. Overall, the structure shows that PI3KC3-C1 and FIP200 interact extensively and intimately. It is striking that the BECN1[BH3] and ATG14–BECN1 zinc-finger regions become ordered and wrap more than halfway around the FIP200 proximal arm and that these regions closely approach ATG13 sites 1 and 2.

## FIP200 scaffolding of ULK1 dimerization

In the same ULK1C:PI3KC3-C1 mixture dataset described above, a particle class of ULK1C showed clear 2:2:2 stoichiometry (Fig. 5a and

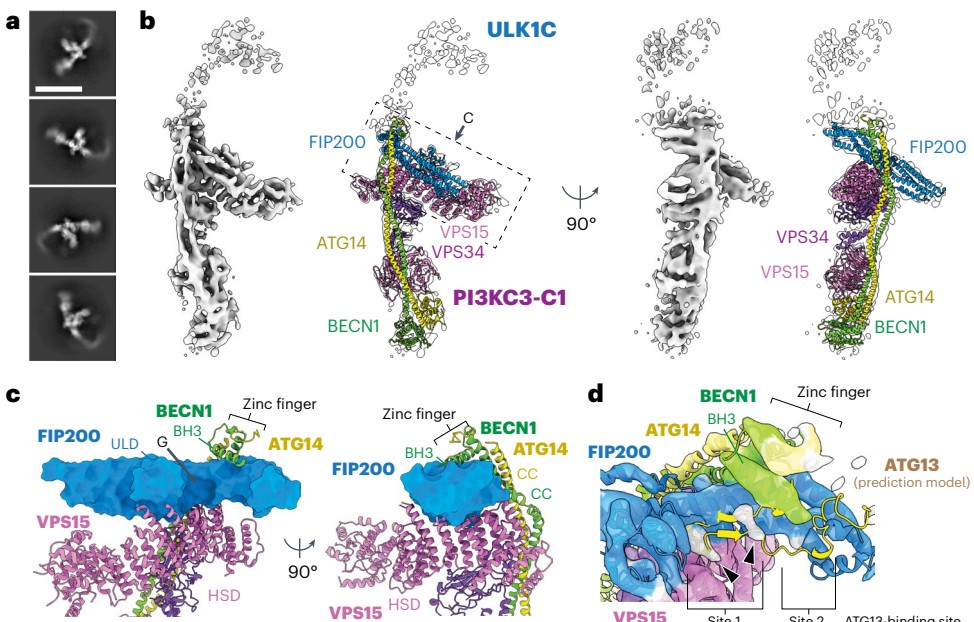

**Fig. 4 | Structure of the ULK1C:PI3KC3-C1 supercomplex. a**, Representative 2D class average of the ULK1C:PI3KC3-C1 supercomplex. Scale bar, 20 nm. **b**, Overview of the EM map and the coordinates of ULK1C:PI3KC3-C1 supercomplex superposed with the EM map contoured at $8\sigma$. Note: only one FIP200 molecule of the ULK1C was modeled (blue). **c**, Close-up view of the FIP200-binding interface of PI3KC3-C1. FIP200 is shown in surface representation and PI3KC3-C1 is shown in ribbon representation. The FIP200$^{ULD}$, VPS15$^{HSD}$, BECN1, CC and zinc-finger domains formed by BECN1–ATG14 are indicated. **d**, Close-up view of the unassigned EM densities at sites 1 and 2. The local refinement map is shown at $22\sigma$. The assigned map is colored according to the molecule (blue, FIP200; pink, VPS15; green, BECN1; yellow, ATG14). The extra densities observed at site 1 are shown in gray and indicated by arrows.

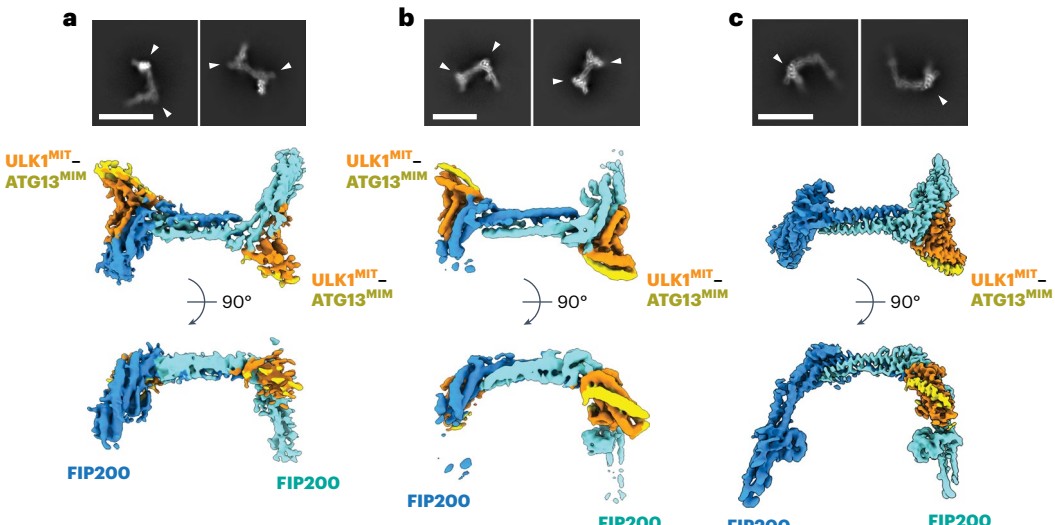

**Fig. 5 | ULK1 dimerization on the FIP200 scaffold. a**, Representative 2D class average and the EM map of the ULK1C (2:2:2) core in the presence of PI3KC3-C1. The ULK1$^{MIT}$:ATG13$^{MIM}$ domains are indicated with arrows. Scale bar, 20 nm. The EM map is contoured at $7\sigma$. **b**, Representative 2D class average and EM map of the ULK1C (2:2:2) core of the ATG13$^{450-517}$ truncation mutant. The EM map is contoured at $14\sigma$. **c**, Repeated 2D class averages and EM map of the ULK1C (2:1:1) core as shown in Fig. 1b,c for comparison. The EM map is contoured at $12\sigma$.

Extended Data Fig. 5c). The presence of the ULK1$^{MIT}$:ATG13$^{MIM}$ heterodimer was confirmed on both FIP200 shoulders, demonstrating that the stoichiometry of the ULK1 complex was altered in the cryo-EM sample by the presence of PI3KC3-C1.

ATG13 site 2 is not involved in the 2:2:2 complex; therefore, we hypothesized that this site might function as a brake to negatively regulate formation of this complex. To test this hypothesis, we truncated ATG13$^{IDR}$ at residue 449 and determined the cryo-EM structure of an ATG13$^{450-517}$-containing version of the ULK1C core in the

absence of PI3KC3-C1. The 2D averages clearly show densities of the ULK1$^{MIT}$:ATG13$^{MIM}$ heterodimer on both sides of the shoulder (Fig. 5b), which was confirmed by 3D reconstruction at 4.46 Å (Extended Data Fig. 5d–h). The stoichiometry was investigated in solution by SEC with in-line multiangle light scattering (MALS) (Extended Data Fig. 8a) and mass photometry (Fig. 6). The ULK1C cores in the presence and absence of the ATG13$^{IDR}$ region displayed masses consistent with 2:1:1 and 2:2:2 complexes, respectively, corroborating the cryo-EM data. Furthermore, the ATG13 F394D;F397D;E403K mutant, designed to disrupt

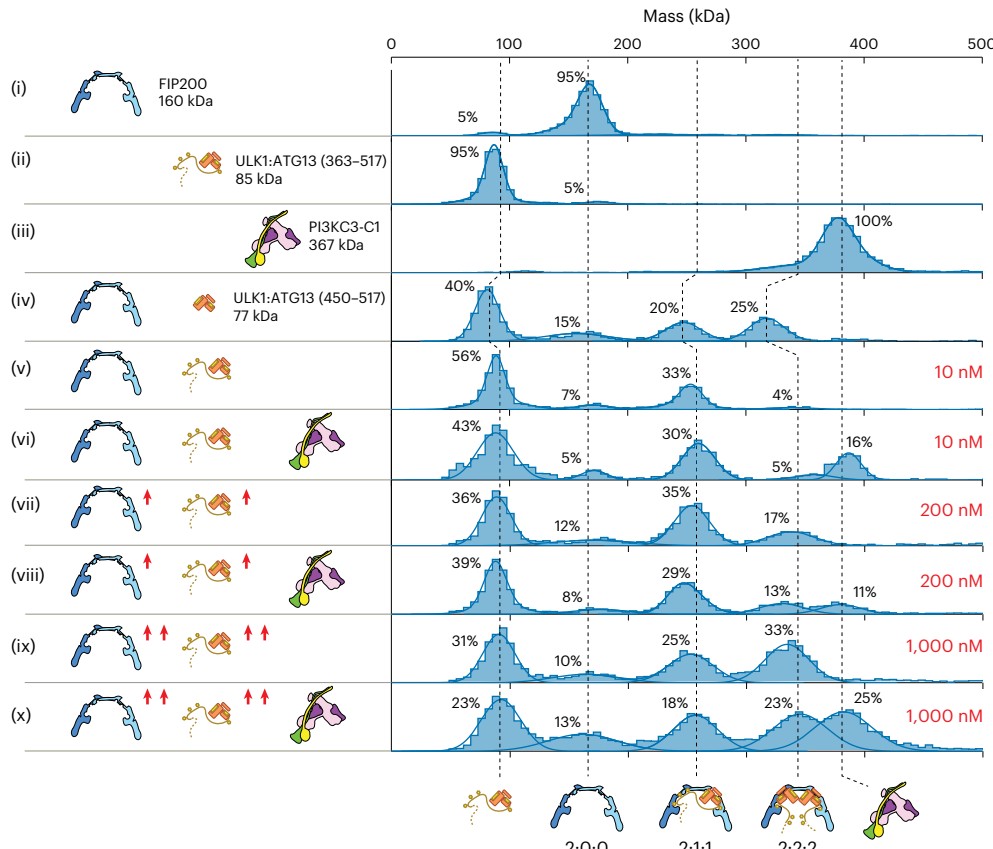

**Fig. 6 | Concentration-dependent stoichiometry shift of ULK1C.** Mass photometry measurements of (i) FIP200[NTD], (ii) ULK1[MIT]:ATG13 (363–517), (iii) full-length PI3KC1-C1, (iv) FIP200[NTD] with truncated ULK1[MIT]:ATG13 (450–517; note: the molecular weight shifted because of the truncation), (v,vi) FIP200[NTD] with ULK1[MIT]:ATG13 (363–517) crosslinked at 10 nM in the absence or presence of 5 nM PI3KC3-C1, (vii,viii) FIP200[NTD] with ULK1[MIT]:ATG13 (363–517) crosslinked at 200 nM in the absence or presence of 100 nM PI3KC3-C1 and (ix,x) FIP200[NTD] with ULK1[MIT]:ATG13 (363–517) at crosslinked 1,000 nM in the absence or

presence of 500 nM PI3KC3-C1. Solid curves represent fits to multiple Gaussian to estimate the average mass and the percentage of each population (Extended Data Table 1). The percentage was calculated from the area of the peaks in the range of 0–500 kDa. The schematic drawing illustrates the setting of each measurement (left) and the predicted assemblies (bottom). Left, the theoretical molecular weights of each assembly are shown next to the diagrams (i–iv). Right, the crosslinking concentrations of FIP200 and ULK1:ATG13 used for each measurement are indicated with red labels (v–x).

site 2, also formed a complex with 2:2:2 stoichiometry (Extended Data Fig. 8b). This observation demonstrates that site 2 of ATG13[IDR] opposes the recruitment of the second ULK1 kinase to the FIP200 dimer.

We sought to determine whether PI3KC3-C1 also promoted the formation of the 2:2:2 complex in solution. Using mass photometry, we observed that the ULK1C core alone at 10 nM concentration formed a 2:1:1 complex (Fig. 6(v)), consistent with the SEC–MALS results. The presence of PI3KC3-C1 did not alter the stoichiometry in solution (Fig. 6(vi)). We hypothesized that PI3KC3-C1 might facilitate the membrane recruitment and increased local concentration of ULK1C in autophagy initiation, which might manifest in a cryo-EM experiment as recruitment and concentration of ULK1 at the air–water interface on EM grids. We found that, upon increasing the concentration from 200 nM to 1 μM, abundant 2:2:2 complex appeared in solution, roughly coequal to the 2:1:1 complex (Fig. 6(ix)). Addition of PI3KC3-C1 did not lead to a further enhancement of the 2:2:2-to-2:1:1 ratio in solution (Fig. 6(x)). This observation shows that mass action alone can drive the 2:1:1-to-2:2:2 state conversion, even in the absence of PI3KC3-C1 and for constructs containing ATG13 site 2.

## Discussion

Despite ULK1C and PI3KC3-C1 being at the heart of human autophagy initiation[35], there have been major gaps in understanding how autophagy is switched on by these two complexes. The available structures of ULK1 complex components have been either too fragmentary

or at inadequate resolution to draw mechanistic conclusions about the integrated regulation of the entire complexes. Beyond the ability of ULK1 to phosphorylate all four subunits of PI3KC3-C1 (refs. 21–24), it has been unclear how the activities of the two complexes are coordinated. Here, we obtained a cryo-EM reconstruction of the ULK1C core that permitted amino-acid-level interpretation. We then found that ULK1C and PI3KC3-C1 form a physical complex mediated by ULK1C FIP200 and PI3KC3-C1 VPS15, BECN1 and ATG14 subunits. The ULK1C core binds full-length PI3KC3-C1 stably enough to yield a cryo-EM reconstruction. Our findings differ from a report indicating that ULK1C binds to a BECN1:ATG14 subcomplex of PI3KC3-C1 with a full-length HORMA domain-containing ATG13 and ATG101 (ref. 36), which can be attributed to the use of the intact PI3KC3-C1 in this study versus the BECN1:ATG14 subcomplex used in the previous study[36]. In yeast, the Atg1 complex (the yeast counterpart of human ULK1C) and PI3KC3-C1 are also reported to be in direct physical contact, which mediates the recruitment of PI3KC3-C1 to the phagophore initiation site in yeast[37]. We note that the ULK1C:PI3KC3-C1 interface is distributed over a large region of the surfaces of these complexes, which has made it challenging to engineer stable mutants that selectively block the interaction. The determinants of binding, however, differ between yeast and humans. Among other differences, Atg38 is important for PI3KC3-C1 recruitment in yeast[37], while we found that the human supercomplex assembles independently of the human Atg38 ortholog NRBF2. The same surface region of VPS15[HSD] binds to both FIP200 and NRBF2

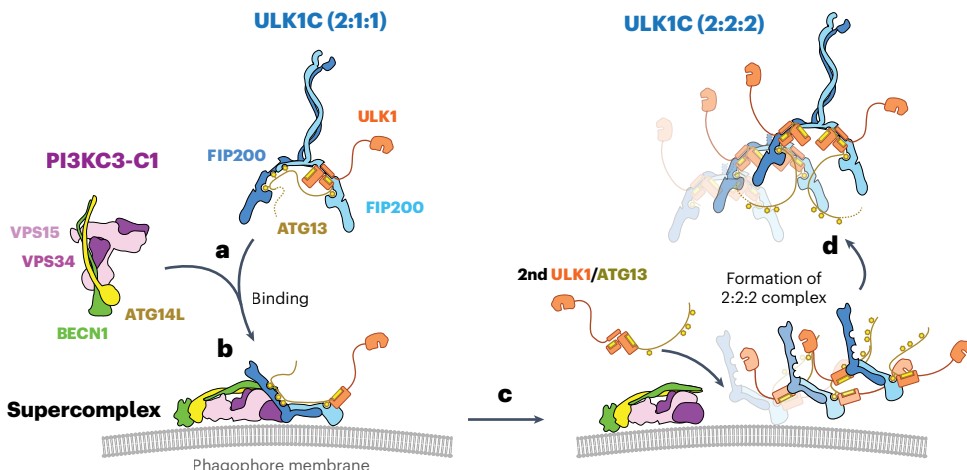

**Fig. 7 | Model for ULK1 activation. a**, Binding and recruitment of the ULK1C to the phagophore membrane by PI3KC3-C1. **b**, Formation of ULK1C:PI3KC3-C1 supercomplex. **c**, Increase in the local concentration of ULK1C facilitated by PI3KC3-C1. **d**, Formation of ULK1 (2:2:2) complex.

(Extended Data Fig. 7c,d)[17]; therefore, the relationship between these interactions remains to be investigated further. The most important conclusions are that ULK1C and PI3KC3-C1 form a physical complex through interactions that involve the autophagy-specific ATG14 subunit of PI3KC3-C1 responsible for autophagy-specific functions.

We previously found that TBK1 contains a scaffold domain that closely resembles the arm of FIP200 involved in binding to VPS15 (Extended Data Fig. 2b) and TBK1 has been shown to bind directly to PI3KC3-C1 (ref. 27). Most but not all autophagy initiation pathways require ULK1 and/or ULK2 (ULK1/2). In OPTN-mediated Parkin-dependent mitophagy, TBK1 replaces the requirement for ULK1/2 (ref. 27). It will be interesting to determine whether TBK1 uses its FIP200 arm-like domain to bypass the need for ULK1/2 in OPTN mitophagy. The role of the VPS15 pseudokinase domain in PI3KC3 function has been a mystery since its initial identification in yeast[38]. We recently elucidated how VPS15 regulates the lipid kinase activity of PI3KC3 (ref. 30). Here, we established a role for VPS15 as the major bridge to FIP200 in ULK1C recruitment. The N-terminal predicted zinc-finger domain of ATG14 is crucial for autophagy initiation and the recruitment of PI3KC3-C1 to initiation sites[5,15]. Yet the molecular function of the ATG14 N-terminal region in the context of PI3KC3-C1 has been elusive and it was not visualized in previous PI3KC3-C1 structures. Here, we found that this region cofolds with the BECN1 NTD and both wrap around the FIP200 proximal arm, consistent with and potentially explaining the function of this region in autophagy initiation. A role for BECN1[BH3] in regulating autophagy through interactions with Bcl-1 is also well established[39–41]. As a limitation, we note that it was not possible to test complex formation by N-terminally truncated forms of ATG14 and BECN1 because of loss of stability. Moreover, the local resolution of the interface in the region involving ATG14 and BECN1[BH3] was insufficient to atomistically model molecular contacts. Our observation that BECN1[BH3] participates in ULK1 supercomplex formation suggests that BECN1[BH3] could have a more direct function in autophagy initiation than previously appreciated.

With respect to the regulation of autophagy initiation, we observed a rearrangement of ULK1C from a 2:1:1 to a 2:2:2 complex. ULK1 kinase activity requires autophosphorylation at T180 (ref. 42), as does its yeast ortholog, Atg1 (ref. 43). Artificially induced dimerization of yeast Atg1 promotes its activation[44,45]. The dimerization and subsequent autophosphorylation of receptor-linked tyrosine kinases (RTKs) is the central paradigm for kinase activation in growth factor signaling[46]. These findings suggest that PI3KC3-C1-regulated and ATG13-regulated ULK1 kinase dimerization could be an autophagic cognate of the dimerization-based RTK activation paradigm. Here, we

observed that the presence of PI3KC3-C1 induced 2:2:2 conversion on cryo-EM grids but not in solution. We established that mass action can drive 2:2:2 conversion in solution in the absence of PI3KC3-C1. We found that ATG13[IDR] site 2 regulates 2:2:2 conversion both on cryo-EM grids and in solution, demonstrating a common molecular mechanism in both settings. In the supercomplex structure, the N-terminal portions of PI3KC3-C1 subunits BECN1 and ATG14 closely approach ATG13[IDR] sites 1 and 2 (Extended Data Fig. 7e,f). ULK1 initiates autophagy in the context of puncta of >30 molecules per cluster that are colocalized with the ATG14 subunit of PI3KC3-C1 (refs. 47,48). However, the small number of molecules involved and the transient nature of autophagy initiation have thus far precluded direct imaging of ULK1 dimerization. The elevated local concentration of ULK1 and associated molecules under conditions of corecruitment and colocalization with PI3KC3-C1 would be expected to help drive formation of the 2:2:2 complex. Taking all of these observations together, the data suggest the existence of regulated ULK1 dimerization on the FIP200 scaffold, which could be a versatile mechanism for ULK1 activation (Fig. 7).

Given the large interface between PI3KC3-C1 and FIP200 and the multiple contacts between ATG13[IDR] and FIP200, it seems likely that various upstream kinases and other signaling inputs could regulate formation of the ULK1C:PI3KC3-C1 supercomplex and the activated 2:2:2 ULK1C. While mammalian target of rapamycin C1 is one candidate to modulate these complexes, these mechanisms are, in principle, general and could be regulated by other kinases such as AMP-activated protein kinase[49] or TBK1 (ref. 50) or by increased local concentration and colocalization driven by selective autophagy cargo receptor clustering[51] or condensate formation[52]. The new activation-related interfaces identified in this study are also suggestive of new concepts for the therapeutic upregulation of autophagy. Thus, the concept of ULK1C:PI3KC3-C1 supercomplex formation at the heart of autophagy initiation puts both the signaling and therapeutic aspects of autophagy initiation into new perspective.

## Online content

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

## Methods

### Plasmid construction

The sequences of all DNAs encoding components of human ULK1C were codon-optimized, synthesized and then subcloned into the pCAG vector. The fusion construct of the ATG13 C-terminal domain with ULK1$^{MIT}$ domain was subcloned with a TwinStrep-Flag (TSF), tobacco etch virus (TEV) cleavage site, ATG13 (363–517 or 450–517) and 5-aa linker (GSDEA) followed by ULK1 (836–1050) into the pCAG vector. Proteins were tagged with GST (glutathione *S*-transferase), MBP (maltose-binding protein) or TSF for affinity purification, pulldown or GSH (glutathione) bead assay. All constructs were verified by DNA sequencing. Details are shown in Supplementary Table 1. A detailed procedure is described at https://doi.org/10.17504/protocols.io.bp2l6x3b5lqe/v1.

### Protein expression and purification

For cryo-EM samples of FIP200$^{NTD}$:ATG13$^{363–517}$–ULK1$^{MIT}$ complexes, pCAG-FIP200$^{NTD}$ (1–640) was cotransfected with pCAG-TSF-ATG13$^{363–517}$-ULK1$^{MIT}$ (836–1050) using the polyethylenimine (PEI) (Polysciences) transfection system. FIP200$^{NTD}$–MBP:ATG13$^{363–517}$–ULK1$^{MIT}$–TSF was coexpressed in HEK293 GnTi$^-$ cells for the FIP200:PI3KC3-C1 cryo-EM study. Cells were transfected at a concentration of $2 \times 10^6$ cells per ml. After 48 h, cells were pelleted at $500g$ for 10 min, washed with PBS once and then stored at −80 °C. Cell pellets were lysed at room temperature for 20 min with lysis buffer (25 mM HEPES pH 7.5, 200 mM NaCl, 2 mM MgCl$_2$, 1 mM TCEP and 10% glycerol) with 5 mM EDTA, 1% Triton X-100 and protease inhibitor cocktail (Thermo Fisher Scientific) before being cleared at 17,000 rpm for 35 min at 4 °C. The clarified supernatant was purified on Strep-Tactin Sepharose resin (IBA Lifesciences) and then eluted in the lysis buffer with 4 mM desthiobiotin (Sigma). After His$_6$–TEV cleavage at 4 °C overnight, samples were concentrated then load onto a Superose 6 Increase 10/300 GL column (Cytiva) in a buffer of 25 mM HEPES pH 7.5, 150 mM NaCl, 1 mM MgCl$_2$ and 1 mM TCEP. A detailed procedure is described at https://doi.org/10.17504/protocols.io.bp2l6x3b5lqe/v1.

The PI3KC3-C1 complex was expressed in HEK293 GnTi$^-$ cells by PEI transfection from codon-optimized pCAG-VPS15-TSF, pCAG-VPS34, pCAG-ATG14 and pCAG-Beclin-1. pCAG-GST-ATG14 or pCAG-mCherry-ATG14 was used in PI3KC3-C1 expression for GST pulldown or microscopy-based GSH bead assay. Cells were transfected at a concentration of $2 \times 10^6$ cells per ml and harvested 48 h after transfection. Pellets were homogenized 20 times by a Pyrex douncer (Corning) in lysis buffer with 25 mM TCEP and proteinase inhibitors (Thermo Fisher Scientific), before adding 10% Triton X-100 stock to a final 1% concentration. After rocking at 4 °C for 1 h, lysates were clarified by centrifugation (43,714$g$ for 40 min at 4 °C) and incubated with Strep-Tactin Sepharose (IBA Lifesciences) at 4 °C overnight. After eluting with lysis buffer and 4 mM desthiobiotin, samples were concentrated and then loaded onto a Superose 6 Increase 10/300 GL column (Cytiva) in 25 mM HEPES pH 7.5, 300 mM NaCl, 1 mM MgCl$_2$ and 25 mM TCEP. FIP200$^{NTD}$ constructs used in Fig. 3a,b were purified using Strep-Tactin Sepharose resin as described above and then loaded onto a Superose 6 Increase 10/300 GL column (Cytiva). ATG101:ATG13N HORMA-related proteins in Fig. 3a,b were purified using Strep resin and a Superdex 200 Increase 10/300 GL column (Cytiva) equilibrated in 25 mM HEPES pH 7.5, 150 mM NaCl, 1 mM MgCl$_2$ and 1 mM TCEP. A detailed procedure is described at https://doi.org/10.17504/protocols.io.5qpvorjezv4o/v1.

### Strep pulldown assay

FIP200$^{NTD}$–TSF wild type (WT) or mutants were cotransfected with GST–ATG13$^{363–517}$ and/or ULK1$^{MIT}$–MBP in 10 ml of HEK293 GnTi$^-$ cells. The cells were harvested 48 h after transfection. The pellets were homogenized in 0.5 ml of lysis buffer, protease inhibitors and 1% Triton X-100 and clarified at 40,000$g$ for 15 min. The lysate was incubated with 30 µl of Strep-Tactin Sepharose resin (IBA Lifesciences) at 4 °C for 3 h. The beads were washed four times and then eluted in 50 µl of lysis buffer and 4 mM desthiobiotin. Then, 18 ml of eluent was mixed with lithium dodecylsulfate (LDS)−β-mercaptoethanol (BME) buffer, heated at 60 °C for 5 min and subjected to SDS–PAGE. The gel was then stained with Coomassie brilliant blue G250. For loading input control, 0.3% lysate per sample was mixed with 1× LDS sample buffer (Life Technologies) containing 100 mM BME, boiled at 95 °C for 5 min and then analyzed with 4–12% Bis-Tris gels (Life Technologies). Gels were electrotransferred to polyvinyldifluoride (PVDF) membranes and immunoblotted with various antibodies. FIP200-NTD–TSF lysate input was visualized using horseradish-peroxidase-conjugated anti-Flag antibody (Abcam, ab49763) and SuperSignal West Femto maximum sensitivity substrate enhanced chemiluminescence kit (Thermo Fisher Scientific, 34095). GST–ATG13c lysate input was analyzed using goat anti-GST antibody (Cytiva, 27457701) and donkey anti-goat IgG (Cy5) (Abcam, ab6566). ULKc–MBP lysate was analyzed using rabbit anti-MBP antibody (Thermo Fisher Scientific, PA1-989) and Alexa Fluor 488 goat anti-rabbit IgG (Invitrogen, A11034). The chemiluminescence or fluorescence signals were detected using a ChemiDocMP Imaging system (Biorad). A detailed procedure is described at https://doi.org/10.17504/protocols.io.3byl4jmpolo5/v1.

### Microscopy-based GSH bead protein–protein interaction assay

A mixture of 1 µM purified GST-tagged protein and 500 nM purified fluorescent protein in a total volume of 70 µl was incubated with 9 µl of preblocked glutathione Sepharose beads (Cytiva) in a reaction buffer containing 25 mM HEPES at pH 7.5, 150 mM NaCl, 1 mM MgCl$_2$ and 1 mM TCEP. After incubation at room temperature for 30 min, samples were mixed with an additional 100 µl of reaction buffer and then transferred to the observation chamber for imaging. Images were acquired on a Nikon A1 confocal microscope with a Nikon Plan APO VC ×20 0.75 numerical aperture ultraviolet microscope objective. Three biological replicates were performed for each experimental condition. A detailed procedure is described at https://doi.org/10.17504/protocols.io.4r3l27xdxg1y/v1.

### SEC−MALS

The purified FIP200$^{NTD}$:ATG13−ULK1$^{MIT}$ complexes were concentrated to 4–5 mg ml$^{-1}$. SEC−MALS experiments were performed using an Agilent 1200 high-performance liquid chromatography system (Agilent Technologies), coupled to a Wyatt DAWN HELEOS-II MALS instrument and a Wyatt Optilab rEX differential refractometer (Wyatt). For chromatographic separation, a WTC-050S5 size-exclusion column (Wyatt) with a 40-µl sample loop was used at a flow rate of 0.3 ml min$^{-1}$ in the buffer of 25 mM HEPES pH 7.5, 200 mM NaCl, 1 mM MgCl$_2$ and 2 mM TCEP. The outputs were analyzed using ASTRA V software (Wyatt). MALS signals, combined with the protein concentration determined by refractive index, were used to calculate the molecular mass of the complex. A detailed procedure is described at https://doi.org/10.17504/protocols.io.j8nlkom4xv5r/v1.

### Sample preparation of FIP200:PI3KC3-C1 complex for cryo-EM

FIP200$^{NTD}$–MBP:ATG13$^{363–517}$–ULK1$^{MIT}$–TSF (final 5 µM) was mixed with PI3KC3-C1–TSF complex at a 1.5:1 molar ratio in a total volume of 200 µl, rocking at 4 °C overnight. On the next day, the sample was incubated with 50 µl of Strep-Tactin Sepharose resin (IBA Lifesciences) at 4 °C for 3 h. After one wash, the beads were eluted in 50 µl of 10 mM D-desthiobiotin with the buffer of 25 mM HEPES pH 7.5, 200 mM NaCl, 1 mM MgCl$_2$ and 10 mM TCEP. A detailed procedure is described at https://doi.org/10.17504/protocols.io.5qpvorjezv4o/v1.

### Sample vitrification and cryo-EM data acquisition

For cryo-EM sample preparation, 3 µl of protein solution was applied onto a grid freshly glow-discharged in PELCO easiGlow system (Ted Pella). In-house graphene grids were prepared from Trivial Transfer

graphene sheets (ACS Material) and Quantifoil R2/1 300-mesh gold (EM Sciences) following a protocol introduced by Ahn et al.[53]. The graphene grids were used for the ULK1C (2:1:1) core sample and holey carbon grids (Quantifoil R1.2/1.3 or R2/1 300-mesh; EM Sciences) were sufficient for the other samples. The samples were vitrified with a Vitrobot cryo-plunger (Thermo Fisher Scientific) in plunging conditions optimized previously. Then, 0.05% (w/v) of $n$-octyl-β-D-glucopyranoside was added in the sample solution as a surfactant before vitrification.

The datasets of the ULK1C (2:1:1) core, the ULK1C:PI3KC3-C1 mixture and the ULK1C:PI3KC3-C1 pulldown were recorded at a 300 kV Titan Krios microscope (Thermo Fisher Scientific) equipped with X-FEG and an energy filter set to a width of 20 eV. Automated data acquisition was achieved using SerialEM[54] on a K3 Summit direct detection camera (Gatan) at a magnification of ×81,000 and a corresponding pixel size of 0.525 Å in super-resolution mode with a defocus range of −0.8 to −2.0 μm. The ULK1C:PI3KC3-C1 pulldown sample was collected on a specimen stage tilted at 0°, 20° and 30°. Image stacks with 50 frames was collected with a total dose of 50 e⁻ per Å². The dataset of the ULK1C (2:2:2) core was recorded on a 200-kV Talos Arctica microscope (Thermo Fisher Scientific) equipped with the K3 Summit camera in super-resolution correlated double-sampling mode. The magnification and the pixel size were ×36,000 and 0.5575 Å in super-resolution mode, respectively. Other details of the dataset collection are summarized in Table 1. A detailed procedure is described at https://doi.org/10.17504/protocols.io.kqdg39rreg25/v1.

### Image processing and 3D reconstruction
The datasets were processed by following the workflow in cryoSPARC[55]. In brief, the super-resolution video stacks were motion-corrected and binned 2× by Fourier cropping using Patch motion correction. Contrast transfer function (CTF) determination was performed using Patch CTF estimation, followed by manual removal of the outlier micrographs on the basis of the estimated defocus and resolution value. Single particles were automatically picked by Topaz[56] using a manually trained model and extracted with a window size 1.5 times larger than the target particle before further binning 2× to 4× to facilitate subsequent processing. A 2D classification was then used to remove obvious junk particles. The initial models were obtained using ab initio reconstruction. In case the reconstruction job failed to give a healthy initial model, the classes displaying high-resolution features in the 2D classification step were selected and used for ab initio reconstruction. Further classification was performed at the 3D level by multiple rounds of heterogeneous refinement until a clean substack was obtained. The particles were re-extracted with the refined coordinates on micrographs at the original 2× bin pixel size and used for homogeneous refinement for multiple rounds until the final resolution converged. To further improve the quality of the map, local refinement was applied to the datasets of the ULK1C (2:1:1) core and the ULK1C:PI3KC3-C1 pulldown sample. Masking areas were decided by 3D Variability or 3D Flex and the masks were created by UCSF ChimeraX[57] and Volume Tools. Each local map was aligned to the consensus map and composed using EMAN2 (ref. 58). The composed maps were then used for model building. The details of data processing are summarized in Table 1. A detailed procedure is described in https://doi.org/10.17504/protocols.io.x54v9d99mg3e/v1.

### Model building, validation and visualization
The in silico models of the ULK1C (2:1:1) core and the full-length PI3KC3-C1 were generated by AlphaFold2 prediction[29]. The resolution of the observed maps enabled amino acid sequence assignment. The primary and secondary structure of the predicted models agreed well with the EM maps. A flexible model fitting using the real-time molecular dynamics simulation-based program ISOLDE[59] implemented in the visualization software UCSF ChimeraX was performed, followed by iterative refinement by using the model-editing software Coot[60] manually and real-space refinement in PHENIX[61] automatically.

The two ULK1C (2:2:2) core models were created by aligning two copies of FIP200$^{NTD}$:ATG13$^{363-517}$:ULK1$^{MIT}$ subtrimeric complex to the FIP200 dimerization domain, followed by flexible model fitting with using ISOLDE in ChimeraX. The side chains were removed because of the moderate resolution of these maps. The ULK1C:PI3KC3-C1 supercomplex coordinates were generated by fitting of the individual structures of the ULK1C (2:1:1) and the PI3KC3-C1 to the cryo-EM map. The side chains were removed. The quality of the models was validated by using the validation tools in PHENIX and the online validation service provided by wwPDB[62,63]. The details of the model quality assessment are summarized in Table 1. All figures and videos were made using UCSF ChimeraX. The detailed procedure is described at https://doi.org/10.17504/protocols.io.j8nlkw77wl5r/v1.

### AlphaFold2 model prediction
Models of the FIP200$^{NTD}$:ATG13$^{MIT}$:ULK$^{MIM}$ core domain were generated using the ColabFold implementation of AlphaFold2 (refs. 29,64–67). The default MMseqs2 (ref. 68) pipeline was used to generate and pair multiple-sequence alignments for each protein complex predicted. We predicted the following complexes: dimeric FIP200 (1–640), ULK1 (828–1050) and ATG13 (363–517); dimeric FIP200 (1–640) and ATG13 (363–450); dimeric FIP200 (1–640), ATG13 (450–517) and ULK (828–1050); monomeric FIP200 (1–640) and ATG13 (363–450); monomeric FIP200 (1–640), ATG13 (450–517) and ULK (828–1050). In each case, the models were assessed on the basis of the global predicted template modeling score (pTM) and the interfacial PTM (iPTM) score, with iPTM > 0.5 meriting further inspection. The predicted contacts involving ATG13, FIP200 and ULK1 were manually inspected in UCSF ChimeraX and the local predicted local distance difference test (pLDDT) score of the contacting residues was used in conjunction with an assessment of the physiological environment of the interfaces to judge the quality of the predicted residue contacts. The highest-scoring local pLDDT interfaces were used for figure making and interpretation of HDX data.

### Generation of KO lines using CRISPR–Cas9
CRISPR guide RNAs (gRNAs) targeting ATG13 (5′-CCGCGAGTTTGAT GCCTTTG-3′ (ref. 69) and 5′-TTGCTTCATGTGTAACCTCTGGG-3′) were cloned into a BbsI-linearized pSpCas9(BB)-2A-GFP vector[70] (a gift from F. Zhang; Addgene, plasmid 48138)[70] using a Gibson cloning kit (New England Biolabs). gRNA-containing constructs were then sequence-verified and transfected into WT HeLa cells with X-tremeGENE 9 (Roche) overnight and GFP-positive single cells were sorted by fluorescence-activated cell sorting into 96-well plates. Single-cell colonies were left to grow and screened by immunoblotting for loss of the targeted gene product. This procedure is described in detail at https://doi.org/10.17504/protocols.io.j8nlkkzo6l5r/v1.

### Cloning and generation of stable cell lines
pBMN-HA-WT ATG13 was previously described (Addgene, 186223)[71]. Complementary DNAs encoding ATG13 mutants (site 2, F394D; M396D;F397D;E403K; site 4, F449D;A452D;F453D;S454A; site M, L495D;L499D; sites 2, 4 and M, all substitutions) were synthesized by Integrated DNA Technologies and cloned into pBMN-HA-C1 (Addgene, 188645) using a Gibson cloning kit (New England Biolabs). Generation of stable cell lines was previously described[72] and the detailed procedure is available at https://doi.org/10.17504/protocols.io.dm6gpjzm8gzp/v1. pMRX-IP-HaloTag7-LC3 (Addgene, 184899) and HA-tagged ATG13 variants were stably expressed in ATG13-KO cells using retroviral transduction.

### Halo assay to quantify starvation induced autophagy
We measured autophagic flux using a previously described method[72,73]. In brief, 350,000–400,000 cells were seeded the day before treatment in six-well plates. Cells were treated with 50 nM TMR ligand (Promega)

**Table 1 | Cryo-EM data collection, refinement and validation statistics**

| | ULK1C core (2:1:1) stoichiometry | ULK1C:PI3KC3-C1 supercomplex | ULK1C core (2:2:2) stoichiometry in PI3KC3-C1 mixture | ULK1C core (2:2:2) stoichiometry ATG13[450–517] mutant |
|---|---|---|---|---|
| | (EMD-40658), (PDB 8SOI) | (EMD-45297), (PDB 9C82) | (EMD-40715), (PDB 8SQZ) | (EMD-40735), (PDB 8SRM) |
| **Data collection and processing** | | | | |
| Magnification | ×81,000 | ×81,000 | ×81,000 | ×36,000 |
| Voltage (kV) | 300 | 300 | 300 | 200 |
| Electron exposure ($e^-$ per Å$^2$) | 50 | 50 | 50 | 50 |
| Defocus range (µm) | −0.8 to −2.0 | −0.8 to −2.0 | −0.8 to −2.0 | −0.8 to −2.0 |
| Pixel size (Å) | 1.05 | 1.05 | 1.05 | 1.115 |
| Symmetry imposed | $C_1$ | $C_1$ | $C_1$ | $C_1$ |
| Images | 8,468 | 589 (0° tilt) 6,973 (20° tilt) 3,740 (30° tilt) | 5,383 | 2,286 |
| Initial particle images (no.) | 2,156,516 | 4,473,551 | 502,985 | 968,884 |
| Final particle images (no.) | 267,719 | 21,937 | 97,987 | 149,409 |
| Map resolution, global (Å) | 4.2 | 8.03 | 5.28 | 4.46 |
| FSC threshold | 0.143 | 0.143 | 0.143 | 0.143 |
| Map resolution range (Å) | 3.48–63.83 | 7.15–23.07 | 4.96–81.03 | 3.77–70.64 |
| **Refinement** | | | | |
| Initial model used (PDB code) | AlphaFold2 model | AlphaFold2 model | 8SOI | 8SOI |
| Model resolution (Å) | 3.7 | 8.7 | 6.8 | 6.5 |
| FSC threshold | 0.5 | 0.5 | 0.5 | 0.5 |
| Model resolution range (Å) | 3.48–22.49 | 7.15–16.90 | 4.98–16.47 | 3.78–11.37 |
| Map sharpening $B$ factor (A$^2$) | 143.2 | 752.0 | 431.6 | 155.4 |
| Model composition | | | | |
| Nonhydrogen atoms | 10,243 | 11,172 | 7,438 | 5,773 |
| Protein residues | 1,276 | 2,254 | 1,498 | 1,163 |
| Ligands | 0 | 0 | 0 | 0 |
| $B$ factors (Å$^2$) | | | | |
| Protein | 126.43 | 1,099.03 | 319.79 | 208.69 |
| Ligands | - | - | - | - |
| Root-mean-square deviations | | | | |
| Bond lengths (Å) | 0.003 | 0.002 | 0.004 | 0.002 |
| Bond angles (°) | 0.598 | 0.607 | 0.633 | 0.405 |
| **Validation** | | | | |
| MolProbity score | 1.7 | 2.48 | 1.95 | 1.51 |
| Clashscore | 9.52 | 30.51 | 8.07 | 4.44 |
| Poor rotamers (%) | 1.5 | - | - | - |
| Ramachandran plot | | | | |
| Favored (%) | 96.75 | 91.17 | 91.41 | 95.87 |
| Allowed (%) | 3.17 | 8.83 | 8.59 | 4.13 |
| Disallowed (%) | 0.008 | 0 | 0 | 0 |

PDB, Protein Data Bank; FSC, Fourier shell correlation.

in growth medium (DMEM with 10% FBS, 4.5 g L$^{-1}$ glucose (Sigma), 1× GlutaMAXTM (Thermo Fisher Scientific), 1× nonessential amino acids (Thermo Fisher Scientific) and 25 mM HEPES. After 15 min, cells were washed three times with 1× Dulbecco's PBS and treated with Earle's balanced salt solution (EBSS; Gibco) for the indicated time periods. Following treatment, samples were analyzed by immunoblotting as described below. This procedure is described at https://doi.org/10.17504/protocols.io.3byl4qexzvo5/v1.

**Immunoblotting**

Following feeding and treatment periods, the cells were washed with ice-cold 1× PBS, harvested using cell scrapers and lysed in lysis buffer (1× LDS sample buffer (Life Technologies) containing 100 mM DTT (Sigma)). Samples were boiled at 99 °C with shaking for 7 min and 25 µg (for Halo assays) and 70 µg (for the remaining blots) of protein per sample was analyzed with 4–12% Bis–Tris gels (Life Technologies) according to the manufacturer's instructions. Gels were electrotransferred to

PVDF membranes and immunoblotted with the indicated antibodies. For quantification, band intensities were measured using ImageLab 5.2.1 (BioRad). Statistical significance was calculated from at least three independent experiments using a two-way analysis of variance (ANOVA). Error bars are the means ± s.d. The procedure for immunoblotting is described in detail at https://doi.org/10.17504/protocols.io.dm6gpjzm8gzp/v1.

## Mass photometry

High-precision coverslips (Azer Scientific) were cleaned by alternating between isopropanol and water three times in a bath sonicator, each 3 min, followed by air drying. The gasket was similarly cleaned with isopropanol and water three times without sonication and then air-dried. A volume of 19 µl of mass photometry buffer (30 mM HEPES pH 7.4, 5 mM MgSO$_4$ and 1 mM EGTA) was added to the well for autofocus. The protein sample was then diluted in the mass photometry buffer in the wells to a concentration of 5–20 nM before data collection. If crosslinking was performed, 1 µl of 0.1% (v/v) glutaraldehyde in mass photometry buffer was added to 9 µl of protein sample and incubated for 10 min. The reaction was then quenched by adding 10 µl of 1 M Tris·HCl pH 7.4. All samples were diluted to 10 nM for the measurement of mass photometry. Protein contrast counts were acquired using a Refeyn TwoMP mass photometer, with three technical replicates. Mass calibration was performed using a standard mix of conalbumin, aldolase and thyroglobulin. Mass photometry profiles were analyzed and fitted to multiple Gaussian peaks, with the mean, s.d. and percentages calculated using DiscoverMP software (Refeyn). This procedure is described at https://doi.org/10.17504/protocols.io.kqdg3keq7v25/v1.

## Reporting summary

Further information on research design is available in the Nature Portfolio Reporting Summary linked to this article.

## Data availability

The cryo-EM maps were deposited to the EM Data Bank (EMDB) under accession codes EMD-40658 (ULK1C (2:1:1) core), EMD-45297 (ULK1C:PI3KC3-C1 supercomplex), EMD-40715 (ULK1C (2:2:2) core in the PI3KC3-C1 mixture) and EMD-40735 (ULK1C (2:2:2) core of the ATG13$^{450-517}$ truncation mutant). The structural coordinates were deposited to the PDB under accession codes 8SOI (ULK1C (2:1:1) core), 9C82 (ULK1C:PI3KC3-C1 supercomplex), 8SQZ (ULK1C (2:2:2) core in the PI3KC3-C1 mixture) and 8SRM (ULK1C (2:2:2) core of the ATG13$^{450-517}$ truncation mutant). Protocols were deposited to protocols.io for plasmid construction (https://doi.org/10.17504/protocols.io.bp2l6x3b5lqe/v1), sample preparation for cryo-EM samples of FIP 200$^{NTD}$:ATG13$^{(363-517)}$–ULK1$^{MIT}$ complexes (https://doi.org/10.17504/protocols.io.e6nvwjxw7lmk/v1), Strep pulldown assay (https://doi.org/10.17504/protocols.io.3byl4jmpolo5/v1), GST pulldown assay (https://doi.org/10.17504/protocols.io.36wgqj2xxvk5/v1), microscopy-based GSH bead protein–protein interaction assay (https://doi.org/10.17504/protocols.io.4r3l27xdxg1y/v1), SEC–MALS (https://doi.org/10.17504/protocols.io.j8nlkom4xv5r/v1), sample preparation of FIP200:PI3KC3-C1 complex for cryo-EM (https://doi.org/10.17504/protocols.io.5qpvorjezv4o/v1), image processing and 3D reconstruction (https://doi.org/10.17504/protocols.io.x54v9d99mg3e/v1), model building, validation and visualization (https://doi.org/10.17504/protocols.io.j8nlkw77wl5r/v1), sample vitrification and cryo-EM data acquisition (https://doi.org/10.17504/protocols.io.kqdg39rreg25/v1), generation of CRISPR constructs (https://doi.org/10.17504/protocols.io.j8nlkkzo6l5r/v1) and mass photometry (https://doi.org/10.17504/protocols.io.kqdg3keq7v25/v1). Raw data files for gel scans were uploaded to Zenodo (https://doi.org/10.5281/zenodo.10056244)[74]. Source data files for mass photometry were uploaded to Zenodo (https://doi.org/10.5281/zenodo.15047085)[75]. Plasmids developed for

this study were deposited at Addgene (complete details in Supplementary Table 1). Data and materials can be obtained from the corresponding authors upon request. Source data are provided with this paper.

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

## Acknowledgements

We thank A. Yokom and X. Shi for contributing to early stages of the project. We thank all members of the J.H.H. lab and D. Fracchiolla and others in the Aligning Science Across Parkinson's (ASAP) team mito911 for advice and discussions. We thank D. Toso, P. Tobias and R. Thakkar for cryo-EM facility support and E. Nogales for use of mass photometry instrumentation. This research was funded by ASAP (ASAP-000350) through the Michael J. Fox Foundation for Parkinson's Research (to M.L. and J.H.H.) and National Institutes of Health (R01 NS134598 to J.H.H. and M.L. and R35GM136414 to A.Y.).

## Author contributions

Conceptualization, J.H.H. Methodology, M.C., X.R., T.N.N., G.K., Y.Z. and A.C. Investigation, M.C., X.R., T.N.N., Y.Z. and A.C. Visualization, M.C. Supervision, M.L., A.Y. and J.H.H. Writing—original draft, M.C. and J.H.H. Writing—review and editing, all authors.

## Competing interests

J.H.H. is a cofounder of Casma Therapeutics and receives research funding from Hoffmann-La Roche. M.L. is a cofounder and member of the scientific advisory board of Automera. The other authors declare no competing interests.

## Additional information

**Extended data** is available for this paper at https://doi.org/10.1038/s41594-025-01557-x.

**Correspondence and requests for materials** should be addressed to James H. Hurley.

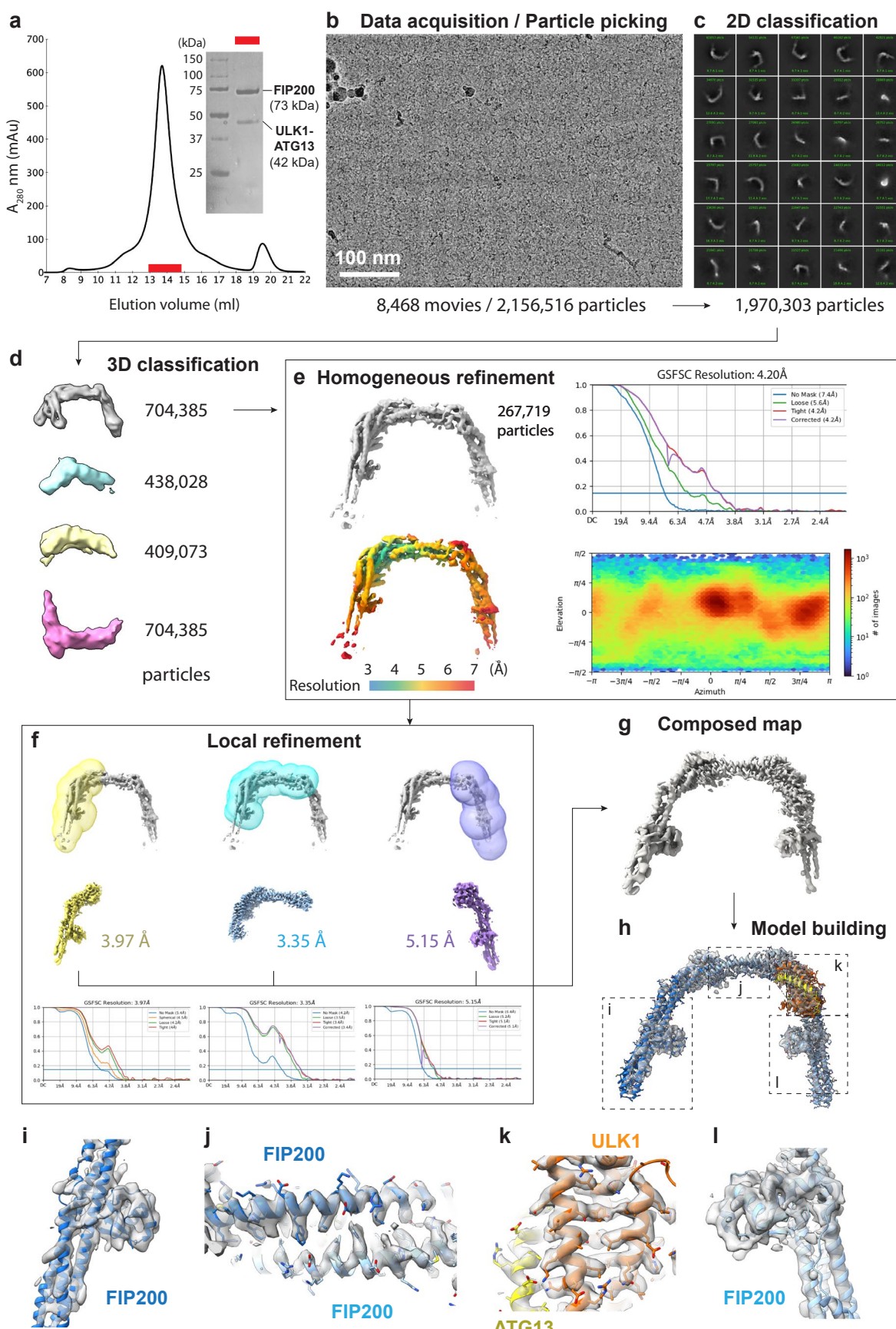

**Extended Data Fig. 1 | See next page for caption.**

**Extended Data Fig. 1 | Cryo-EM sample preparation, image acquisition, data processing, and model building of ULK1C (2:1:1) core. a**, Size exclusion chromatography (SEC) profile of the ULK1C core. The inset shows an SDS-PAGE of the peak (red bar). **b**, A representative cryo-EM micrograph of the ULK1C core. **c**, Representative 2D class averages. **d**, Result of the first round of 3D classification. **e**, Global refinement of the final substack cleaned by multi-rounds of 3D classification. The FSC, local resolution, and angular distribution are shown accordingly. **f**, Masks for local refinement and the corresponding maps and FSCs. **g**, Final composed map for model building. **h**, Overview of the model building. The map is contoured at 12σ. **i** to **l**, Close-up views of the map superposed with the models. Each position is indicated in **h**.

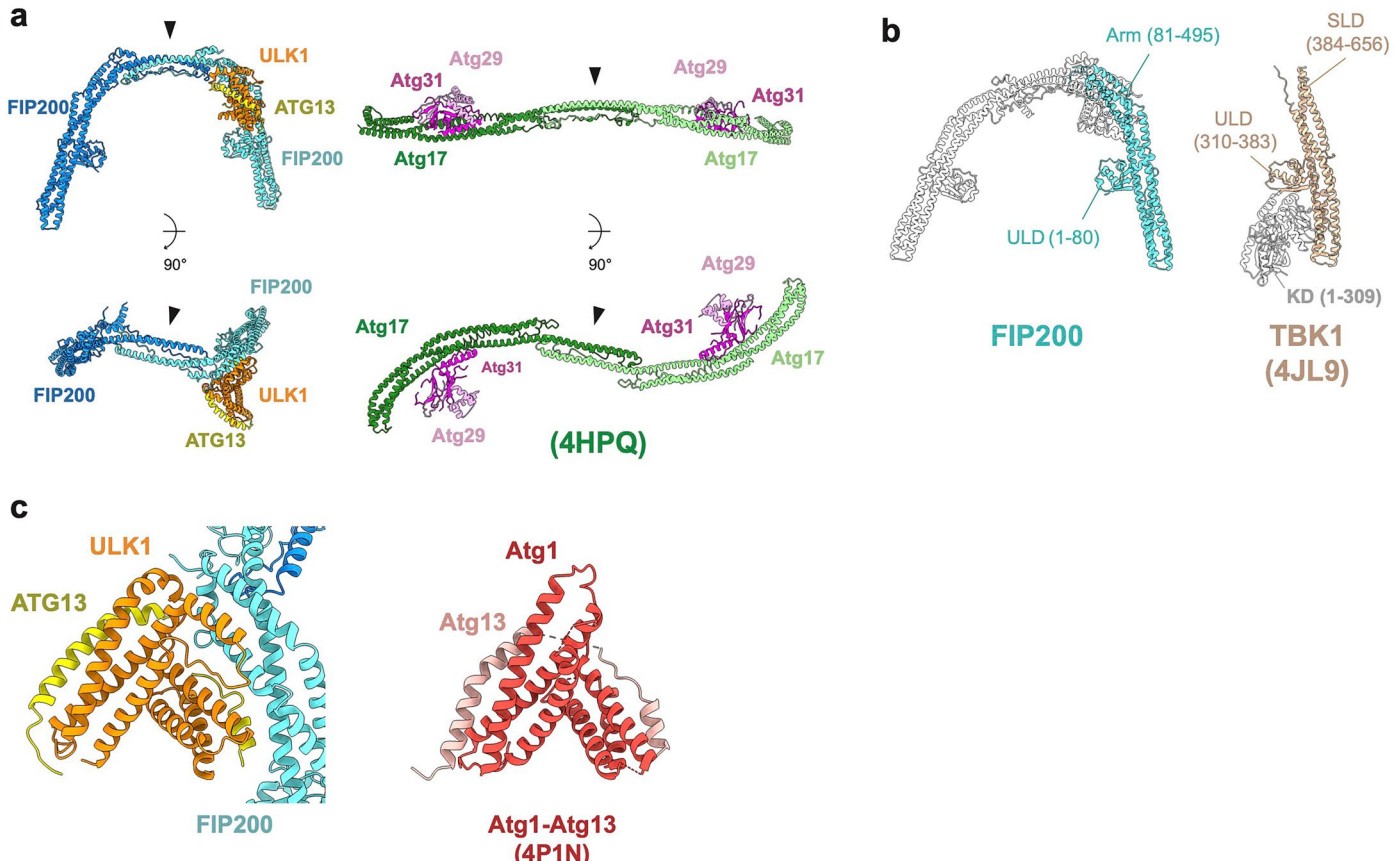

**Extended Data Fig. 2 | Structural comparison of the ULK1C (2:1:1) core with related proteins. a**, Structural comparison of the FIP200 with its yeast homolog Atg17 (PDBID: 4HPQ). The dimerization domains are indicated with arrows.

**b**, Structural comparison of the FIP200 with TBK1 (PDBID: 4JL9). **c**, Structural comparison of the ULK1-ATG13 heterodimer with its yeast homolog Atg1-Atg13 (PDBID: 4P1N).

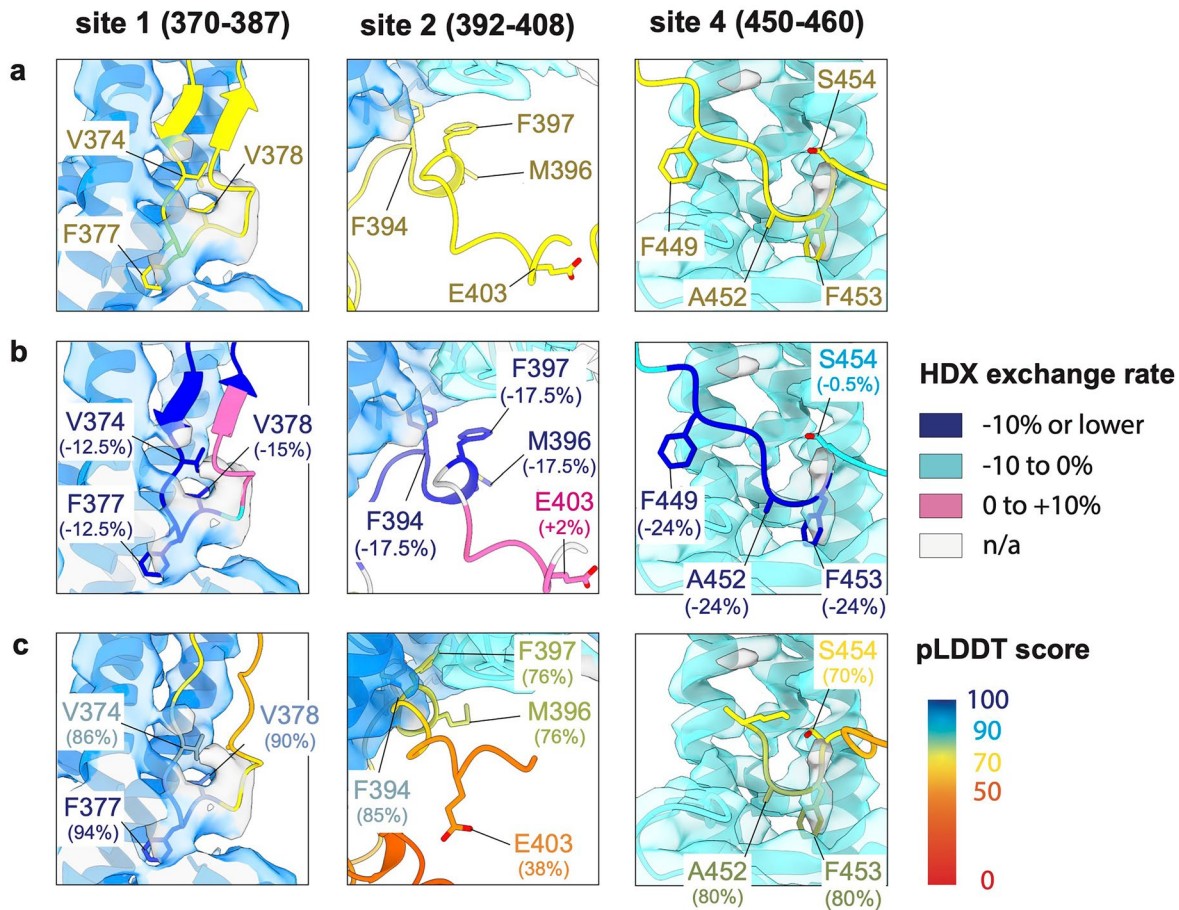

**Extended Data Fig. 3 | Assignment of the ATG13 binding sites. a,** Close up views of the binding site 1, 2, and 4 of the ULK1C core (2:1:1). The map is contoured at 11σ. The AlphaFold2 prediction model was created using dimeric FIP200(1-640), ULK1(828-1050), and ATG13(363-517), and superposed on the cryo-EM structure by aligning the FIP200 molecule. ATG13 is shown in yellow, while the other molecule of the predicted model are not shown. The key binding residues are indicated. **b,** Same close up views of the binding site 1, 2, and 4. The ATG13 molecule is colored based on the HDX exchange rate (-10% or lower: blue; -10% to 0%: cyan; 0% to +10%: pink; n/a: gray) reported previously[19]. The HDX exchange rate for each key residue is shown. **c,** Same close up views of the binding site 1, 2, and 4. For optimizing the predicted local distance difference test (pLDDT) score, AlphaFold2 prediction models were created using monomeric FIP200(1-640) with ATG13(363-450) (left and middle), and monomeric FIP200(1-640) with ATG13(450-517) (right). The ATG13 molecule is colored based on the pLDDT score. (100%: blue; 90%: cyan; 70%: yellow; 50%: orange; 0%: red). The pLDDT score for each key residue is shown.

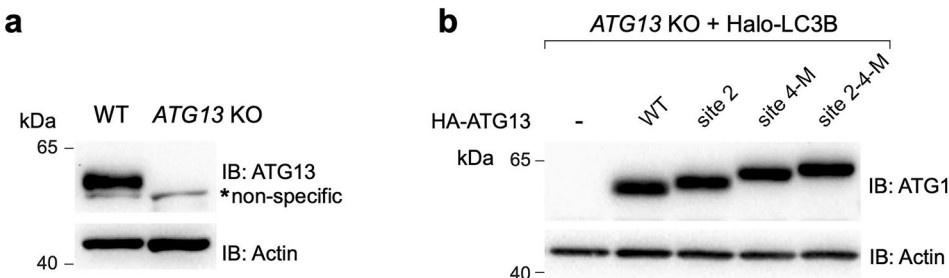

**Extended Data Fig. 4 | Generation of knockout and rescue cell lines.**
**a**, Cell lysates from WT and *ATG13* KO HeLa cells were analyzed by immunoblotting with indicated antibodies. **b**, Cell lysates from *ATG13* KO cells expressing Halo-LC3B without rescue or rescued with indicated versions of HA-ATG13 were immunoblotted with anti-ATG13 and anti-Actin antibodies. These experiments were not repeated because they were designed to confirm the expression of proteins.

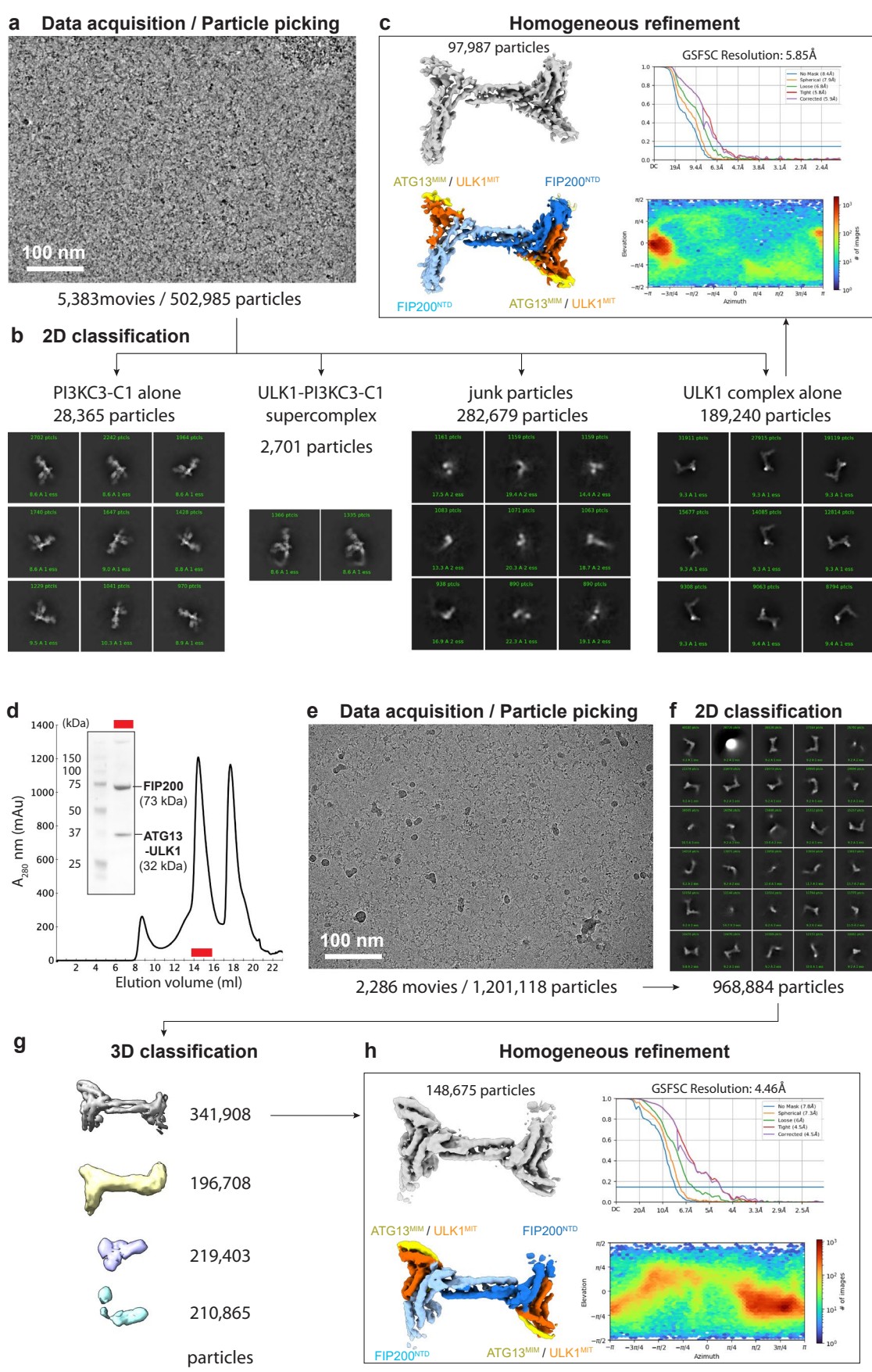

**Extended Data Fig. 5 | See next page for caption.**

**Extended Data Fig. 5 | Structural determination of the ULK1C (2:2:2) core in the PI3KC3-C1 mixture and the ATG13$^{450-517}$ truncation mutant. a**, A representative cryo-EM micrograph of the ULK1C:PI3KC3-C1 mixture sample. **b**, Representative 2D class averages of four subpopulations of the dataset. From left: PI3KC3-C1 alone, ULK1C:PI3KC3-C1 supercomplex, junk particles, and ULK1C alone. **c**, Result of the homogeneous refinement of the ULK1C alone substack. The FSC, local resolution, and angular distribution are shown accordingly. The EM map is contoured at 7σ. **d**, Size exclusion chromatography (SEC) profile of the ULK1C (2:2:2) core of ATG13$^{450-517}$ truncation mutant. The inset shows an SDS-PAGE of the peak (red bar). **e**, A representative cryo-EM micrograph of the ULK1C core containing the ATG13$^{450-517}$ truncation mutant. **f**, Representative 2D class averages. **g**, Result of the first round of 3D classification. **h**, Global refinement of the final substack. The FSC, local resolution, and angular distribution are shown accordingly.

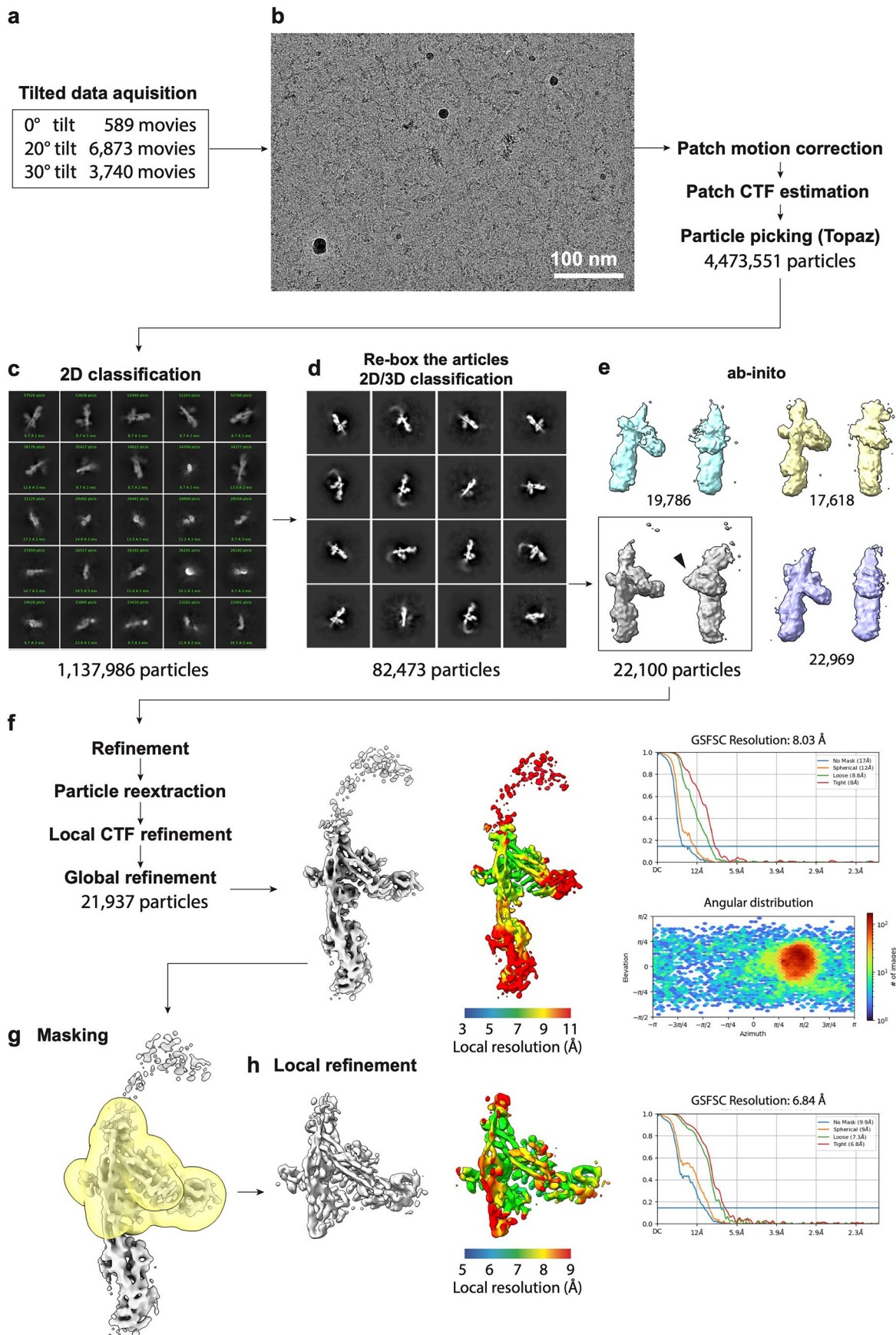

**Extended Data Fig. 6 | See next page for caption.**

**Extended Data Fig. 6 | Structural determination of the ULK1C:PI3KC3-C1 supercomplex. a**, Dataset acquisition. Three data sets were collected at 0°, 20°, and 30° tilted specimen stage. The number of movies collected for each angle is shown respectively. **b**, A representative cryo-EM micrograph is shown on the left. Scale bar: 100 nm. **c**, Representative 2D class averages of the first round of 2D classification. **d**, Representative 2D class averages of the intermediate particle stack, re-extracted with a larger box size, and sorted by multiple rounds of 2D and 3D classification. **e**, The initial model created from the particle stack. The black arrow indicates the EM density of FIP200. **f**, Result of the homogeneous refinement. The local resolution, FSC, and angular distribution are shown accordingly. The map is displayed at a contour level of 8σ. **g** and **h**, Result of the local refinement. The mask for local refinement, local resolution, FSC, and angular distribution are shown accordingly. The map is displayed at a contour level of 17σ.

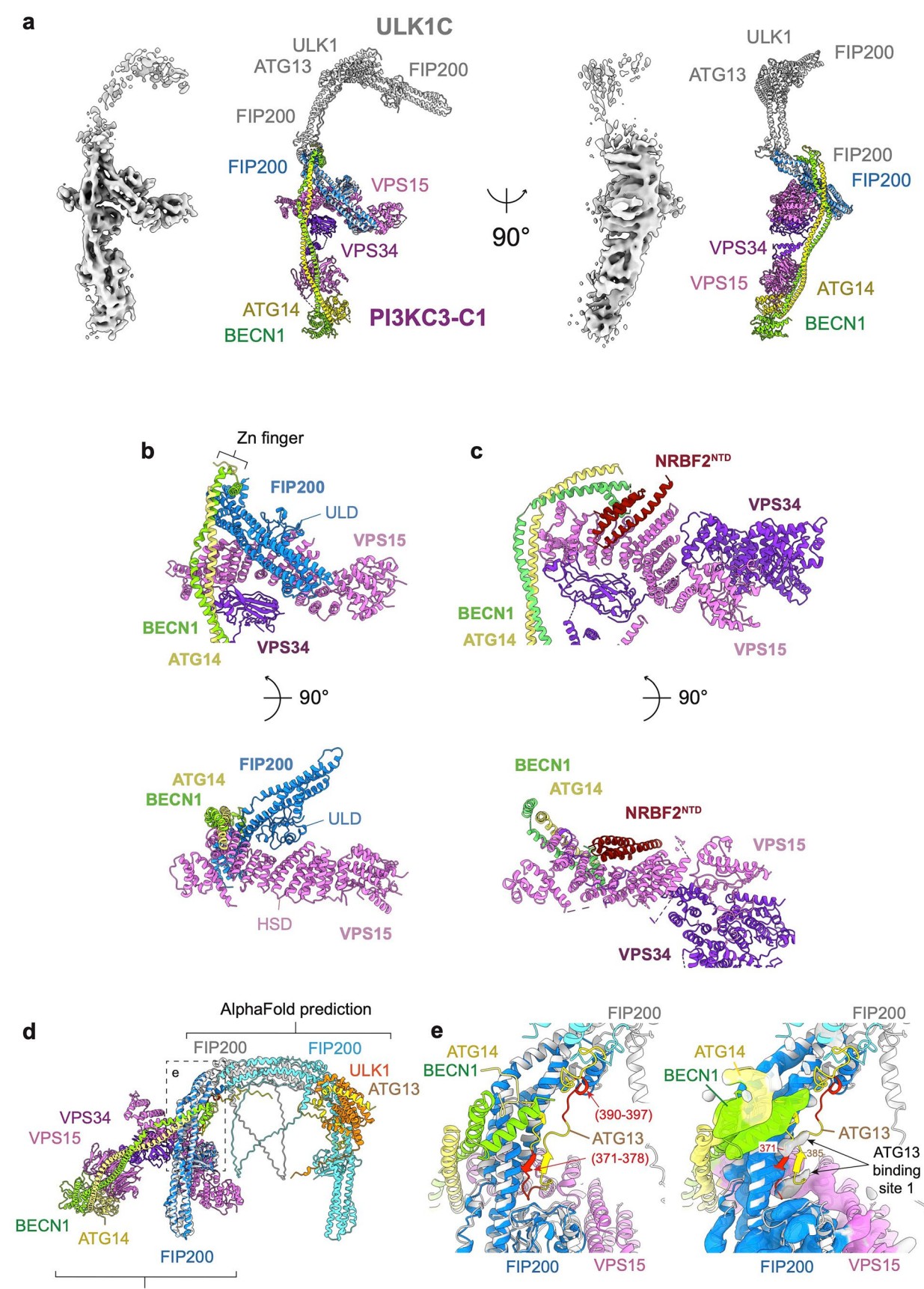

**Extended Data Fig. 7 | See next page for caption.**

**Extended Data Fig. 7 | Characterization of the ULK1C:PI3KC3-C1 supercomplex structure. a**, Overview of the ULK1C:PI3KC3-C1 supercomplex. The cryo-EM structure of the ULK1C core (2:1:1) determined in this study is superposed onto the FIP200 of the supercomplex and colored in gray. **b**, Close up view of the interface between the FIP200 and PI3KC3-C1. The ubiquitin like domain (ULD) of FIP200 and the Zn finger motif are indicated. **c**, Comparison of the VPS15$^{HSD}$ binding sites in binding with NRBF2 (red)[17]. **d**, Superposition of the ULK1C core (2:1:1) prediction model onto the supercomplex cryo-EM structure. The FIP200 molecule of the prediction model (gray) is superposed to the FIP200 molecule of the supercomplex (blue). **e**, Close up view of the FIP200 site 1 and 2. The FIP200 interacting regions of ATG13 (371-378, 390-397), identified with HDX in the previous study[19] are colored red. (Right) The same view with showing the local refinement EM map at 22.5σ. The two extra EM maps observed at site 1 correspond to the predicted ATG13 region 371-385.

**a**

### FIP200^NTD-ULK1^MIT-ATG13^450-517 (2:2:2 stoichiometry)

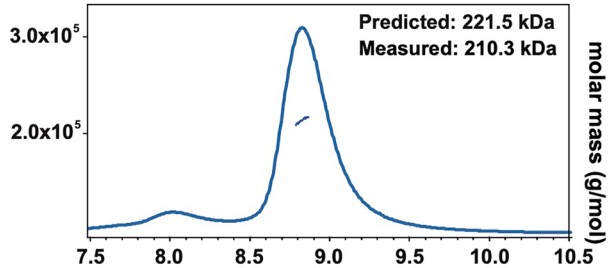
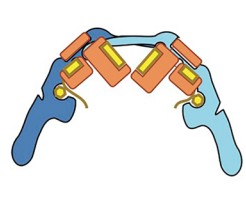

Predicted: 221.5 kDa
Measured: 210.3 kDa

**b**

### FIP200^NTD-ULK1^MIT-ATG13^Site2 Mut (2:2:2 stoichiometry)

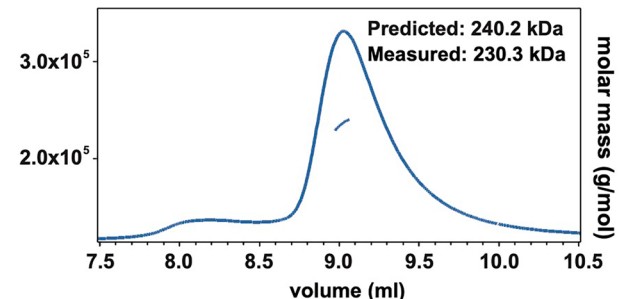
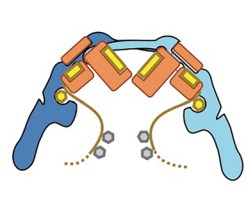

Predicted: 240.2 kDa
Measured: 230.3 kDa

**c**

### FIP200^NTD-ULK1^MIT-ATG13^363-517 (2:1:1 stoichiometry)

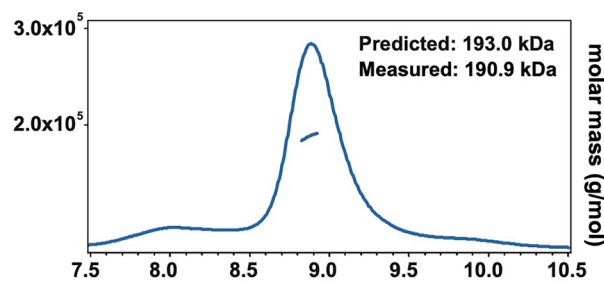
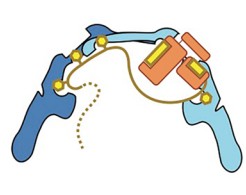

Predicted: 193.0 kDa
Measured: 190.9 kDa

**Extended Data Fig. 8 | ATG13^IDR regulates the 2:2:2 complex formation of the ULK1C. a**, SEC-MALS result of the ULK1C core containing truncated ATG13 variant (ATG13^450-517), **b**, SEC-MALS result of the ULK1C core containing site 2 mutated ATG13 variant (ATG13^363-517 with mutations F394D/F397D/E403K), **c**, SEC-MALS result of the ULK1C core containing wild type ATG13 fragment (ATG13^363-517). The predicted and measured molecular weight for each measurement are shown. The schematic drawing illustrates the predicted assembly of each condition.

**Extended Data Table 1 | Theoretical and measured mass in mass photometry analysis**

| Measurement | Sample | Complex | Expected (kDa) | Measured (kDa) | % |
|---|---|---|---|---|---|
| i) | FIP200n | FIP200n monomer | 80 | 85 | 5 |
| | | FIP200n dimer | 160 | 166 | 95 |
| ii) | MBP-ATG13c-ULK1c | ATG13-ULK1 monomer | 85 | 87 | 95 |
| | | ATG13-ULK1 dimer | 170 | 174 | 5 |
| iii) | PI3KC3-C1 | PI3KC3-C1 | 367 | 377 | 100 |
| iv) | FIP200n + MBP-ATG13(450-517)-ULKc | ATG13-ULK1 monomer | 77 | 81 | 40 |
| | | FIP200n dimer | 160 | 157 | 15 |
| | | ULK1C 2:1:1 complex | 237 | 247 | 20 |
| | | ULK1C 2:2:2 complex | 314 | 318 | 25 |
| v) | FIP200n + MBP-ATG13c-ULK1c Crosslinked at 10 nM | ATG13-ULK1 monomer | 85 | 89 | 56 |
| | | FIP200n dimer | 160 | 170 | 7 |
| | | ULK1C 2:1:1 complex | 245 | 252 | 33 |
| | | ULK1C 2:2:2 complex | 330 | 338 | 4 |
| vi) | FIP200n + MBP-ATG13c-ULK1c + PI3KC3-C1, Crosslinked at 10 nM | ATG13-ULK1 monomer | 85 | 88 | 43 |
| | | FIP200n dimer | 160 | 171 | 5 |
| | | ULK1C 2:1:1 complex | 245 | 260 | 30 |
| | | ULK1C 2:2:2 complex | 330 | 358 | 5 |
| | | PI3KC3-C1 | 367 | 387 | 16 |
| vii) | FIP200n + MBP-ATG13c-ULK1c, Crosslinked at 200nM | ATG13-ULK1 monomer | 85 | 89 | 36 |
| | | FIP200n dimer | 160 | 170 | 12 |
| | | ULK1C 2:1:1 complex | 245 | 254 | 35 |
| | | ULK1C 2:2:2 complex | 330 | 338 | 17 |
| viii) | FIP200n + MBP-ATG13c-ULK1c + PI3K-C1, Crosslinked at 200nM | ATG13-ULK1 monomer | 85 | 89 | 39 |
| | | FIP200n dimer | 160 | 177 | 8 |
| | | ULK1C 2:1:1 complex | 245 | 249 | 29 |
| | | ULK1C 2:2:2 complex | 330 | 330 | 13 |
| | | PI3KC3-C1-C1 | 367 | 380 | 11 |
| ix) | FIP200n + MBP-ATG13c-ULK1c, Crosslinked at 1000nM | ATG13-ULK1 monomer | 85 | 91 | 31 |
| | | FIP200n dimer | 160 | 164 | 10 |
| | | ULK1C 2:1:1 complex | 245 | 254 | 25 |
| | | ULK1C 2:2:2 complex | 330 | 335 | 33 |
| x) | FIP200n + MBP-ATG13c-ULK1c + PI3KC3-C1, Crosslinked at 1000nM | ATG13-ULK1 monomer | 85 | 93 | 23 |
| | | FIP200n dimer | 160 | 162 | 13 |
| | | ULK1C 2:1:1 complex | 245 | 257 | 18 |
| | | ULK1C 2:2:2 complex | 330 | 346 | 23 |
| | | PI3KC3-C1 | 367 | 383 | 25 |

# Reporting Summary

## Statistics

For all statistical analyses, confirm that the following items are present in the figure legend, table legend, main text, or Methods section.

| n/a | Confirmed | |
|---|---|---|
| ☐ | ☒ | The exact sample size (*n*) for each experimental group/condition, given as a discrete number and unit of measurement |
| ☐ | ☒ | A statement on whether measurements were taken from distinct samples or whether the same sample was measured repeatedly |
| ☒ | ☐ | The statistical test(s) used AND whether they are one- or two-sided<br>*Only common tests should be described solely by name; describe more complex techniques in the Methods section.* |
| ☒ | ☐ | A description of all covariates tested |
| ☒ | ☐ | A description of any assumptions or corrections, such as tests of normality and adjustment for multiple comparisons |
| ☐ | ☒ | A full description of the statistical parameters including central tendency (e.g. means) or other basic estimates (e.g. regression coefficient) AND variation (e.g. standard deviation) or associated estimates of uncertainty (e.g. confidence intervals) |
| ☐ | ☒ | For null hypothesis testing, the test statistic (e.g. *F*, *t*, *r*) with confidence intervals, effect sizes, degrees of freedom and *P* value noted<br>*Give P values as exact values whenever suitable.* |
| ☒ | ☐ | For Bayesian analysis, information on the choice of priors and Markov chain Monte Carlo settings |
| ☒ | ☐ | For hierarchical and complex designs, identification of the appropriate level for tests and full reporting of outcomes |
| ☒ | ☐ | Estimates of effect sizes (e.g. Cohen's *d*, Pearson's *r*), indicating how they were calculated |

*Our web collection on statistics for biologists contains articles on many of the points above.*

## Software and code

Policy information about availability of computer code

| | |
|---|---|
| Data collection | SerialEM (ver. 4.0.20), ASTRA (ver. 5.3.4), ImageLab (ver. 5.2.1, DiscoverMP (ver. 2024R2) |
| Data analysis | CryoSPARC (v4.1), Topaz (0.2.5a), ChimeraX (ver 1.5, EMAN2 (ver 2.0), ISOLDE (ver 1.5), Phenix (1.20.1,) AlphaFold (Ver. 2), coot (ver. 0.9.8), ColabFold (ver 1.3.0), MMseqs2 (ver. 13-45111). |

For manuscripts utilizing custom algorithms or software that are central to the research but not yet described in published literature, software must be made available to editors and reviewers. We strongly encourage code deposition in a community repository (e.g. GitHub). See the Nature Portfolio guidelines for submitting code & software for further information.

## Data

Policy information about availability of data

All manuscripts must include a data availability statement. This statement should provide the following information, where applicable:
- Accession codes, unique identifiers, or web links for publicly available datasets
- A description of any restrictions on data availability
- For clinical datasets or third party data, please ensure that the statement adheres to our policy

The cryo-EM maps were deposited in the Electron Microscopy Data Bank (EMDB) under accession codes EMD-40658 (ULK1C (2:1:1) core), EMD-45297 (ULK1C:PI3KC3-C1 supercomplex), EMD-40715 (ULK1C (2:2:2) core in the PI3KC3-C1 mixture), and EMD-40735 (ULK1C (2:2:2) core of the ATG13450-517 truncation mutant). The structural coordinates were deposited in the Protein Data Bank (PDB) under accession codes PDBID:8SOI (ULK1C (2:1:1) core), PDBID:9C82

(ULK1C:PI3KC3-C1 supercomplex), PDBID:8SQZ (ULK1C (2:2:2) core in the PI3KC3-C1 mixture), and PDBID:8SRM (ULK1C (2:2:2) core of the ATG13450-517 truncation mutant). Protocols were deposited in protocols.io (Plasmid construction (DOI: 10.17504/protocols.io.bp2l6x3b5lqe/v1), Sample preparation for cryo-EM samples of FIP200NTD:ATG13(363-517)-ULK1MIT complexes (DOI: 10.17504/protocols.io.e6nvwjxw7lmk/v1), Strep pull-down assay (DOI: 10.17504/protocols.io.3byl4jmpolo5/v1), GST pull down assay (DOI: 10.17504/protocols.io.36wgqj2xxvk5/v1), Microscopy-based GSH bead protein-protein interaction assay (DOI: 10.17504/protocols.io.4r3l27xdxg1y/v1), Size Exclusion Chromatography with Multiangle Light Scattering (SEC-MALS) (DOI: 10.17504/protocols.io.j8nlkom4xv5r/v1), Sample preparation of FIP200:PI3KC3-C1 complex for cryo-EM (DOI: 10.17504/protocols.io.5qpvorjezv4o/v1), Image processing and 3D reconstruction (DOI: 10.17504/protocols.io.x54v9d99mg3e/v1), Model building, validation, and visualization (DOI: 10.17504/protocols.io.j8nlkw77wl5r/v1), Sample vitrification and cryo-EM data acquisition (DOI: 10.17504/protocols.io.kqdg39rreg25/v1)). Generation of CRISPR constructs (DOI: 10.17504/protocols.io.j8nlkkzo6l5r/v1), Mass photometry (DOI: 10.17504/protocols.io.kqdg3keq7v25/v2). Raw data files for gel scans were uploaded to Zenodo (DOI: 10.5281/zenodo.10056244). Source data files for mass photometry were uploaded to Zenodo (DOI: 10.5281/zenodo.15047085). Plasmids developed for this study were deposited at Addgene.org. See Extended Data Table 1 for complete details.

# Research involving human participants, their data, or biological material

Policy information about studies with human participants or human data. See also policy information about sex, gender (identity/presentation), and sexual orientation and race, ethnicity and racism.

| | |
|---|---|
| Reporting on sex and gender | N/A |
| Reporting on race, ethnicity, or other socially relevant groupings | N/A |
| Population characteristics | N/A |
| Recruitment | N/A |
| Ethics oversight | N/A |

Note that full information on the approval of the study protocol must also be provided in the manuscript.

# Field-specific reporting

Please select the one below that is the best fit for your research. If you are not sure, read the appropriate sections before making your selection.

☒ Life sciences ☐ Behavioural & social sciences ☐ Ecological, evolutionary & environmental sciences

For a reference copy of the document with all sections, see nature.com/documents/nr-reporting-summary-flat.pdf

# Life sciences study design

All studies must disclose on these points even when the disclosure is negative.

| | |
|---|---|
| Sample size | For cryo-EM, enough particles were collected to achieve necessary resolution to answer the biological question, as is standard in the field. Cell experiments were performed at least as three replicates as stated in the figure legend, according to current practices in the field. Statistical analysis was performed on experiments for which the sample size included at least 3 biological replicates. Sample sizes were based on previous experience and current standards in the field. |
| Data exclusions | No data were excluded from analyses |
| Replication | Each cryo-EM grid square was collected on once because the holes can only be collected on once. This is standard in the field. Each PAGE gel was repeated two to four times, which was mentioned in figure legends individually. Cell experiments from Figures 2i-2l were repeated 3 times as provided in the figure legend. |
| Randomization | Randomization is not applicable to this study. |
| Blinding | Blinding is not relevant for this study because the experimental conditions must be known to interpret the results. |

# Reporting for specific materials, systems and methods

We require information from authors about some types of materials, experimental systems and methods used in many studies. Here, indicate whether each material, system or method listed is relevant to your study. If you are not sure if a list item applies to your research, read the appropriate section before selecting a response.

# Materials & experimental systems

| n/a | Involved in the study |
|---|---|
| ☐ | ☒ Antibodies |
| ☐ | ☒ Eukaryotic cell lines |
| ☒ | ☐ Palaeontology and archaeology |
| ☒ | ☐ Animals and other organisms |
| ☒ | ☐ Clinical data |
| ☒ | ☐ Dual use research of concern |
| ☒ | ☐ Plants |

# Methods

| n/a | Involved in the study |
|---|---|
| ☒ | ☐ ChIP-seq |
| ☒ | ☐ Flow cytometry |
| ☒ | ☐ MRI-based neuroimaging |

# Antibodies

| | |
|---|---|
| Antibodies used | Anti-VCP (Cell Signaling Technology, Cat# 2649, RRID:AB_2214629, 1:1000), Mouse anti-Actin (Cell Signaling Technology, Cat# 4967S, RRID: AB_330288, 1:1000), Rabbit anti-ATG13 (Cell Signaling Technology, Cat# 7613, RRID: AB_10827645, 1:1000), Rabbit anti-ATG14(S29) (Cell Signaling Technology, Cat# 92340S, RRID: AB_2800182, 1:1000), Rabbit anti-ATG14 (Cell Signaling Technology, Cat# 96752S, RRID: AB_2737056, 1:1000), Rabbit anti-FIP200 (Cell Signaling Technology, Cat# 12436S, RRID: AB_2797913, 1:1000), HRP anti-Flag antibody (Abcam, Cat# b49763, RRID:AB_869428, 1:1000), Donkey anti-goat IgG (Cy5) (Abcam, Cat# ab6566, RRID:AB_955056, 1:300), Anti-GST Antibody (Cytiva, Washington DC, Cat# 27457701, RRID:AB_771432, 1:2000), Rabbit anti-MBP antibody (Thermo Fisher Scientific, Cat# PA1-989, RRID:AB_559988, 1:10000), Alexa Fluor 488 goat anti-Riabbit IgG (Thermo Fisher Scientific, Cat# A11034, RRID:AB_2576217, 1:500), Anti-HALO (Promega, Cat# G9211, RRID:AB_2688011, 1:1000). |
| Validation | For all antibodies used in this study were commercially available. Validation statement for each primary antibody is provided on the manufacture's website: Anti-VCP (Cell Signaling Technology, Cat# 2649): https://www.cellsignal.com/products/primary-antibodies/vcp-7f3-rabbit-mab/2649?srsltid=AfmBOoq4tLBfkpHlr9kvTEwdz8YtTlRJLL4P6SwY0C-ZnwUn07MJbP6l, Validated for WB and cited by 16 publications. Mouse anti-Actin (Cell Signaling Technology, Cat# 4967S): https://www.cellsignal.com/products/primary-antibodies/b-actin-antibody/4967?srsltid=AfmBOopLogEoaBCdOcHv666-eP9SmxBvAxe5V_Q4xTWOAb5PosGBlyqD. Validated for WB and cited by 3824 publications. Rabbit anti-ATG13 (Cell Signaling Technology, Cat# 7613): https://www.cellsignal.com/products/primary-antibodies/atg4a-d62c10-rabbit-mab/7613?srsltid=AfmBOor204sYTb3pG87Rag-1Wtwigzxg8OCIjO_D7-g0EpoyJTd_ZuEk. Validated for WB, IP and cited by 19 publications. Rabbit anti-ATG14(S29) (Cell Signaling Technology, Cat# 92340S): https://www.cellsignal.com/products/primary-antibodies/phospho-atg14-ser29-d4b8m-rabbit-mab/92340?srsltid=AfmBOoo3AW_a5y7F5Z0XDmTIG4kqQxJ4sFQvliD3ULKI3rbIhXYs7nLd. Validated for WB, IF, F and cited by 23 publications. Rabbit anti-ATG14 (Cell Signaling Technology, Cat# 96752S): https://www.cellsignal.com/products/primary-antibodies/atg14-d1a1n-rabbit-mab/96752?srsltid=AfmBOor7DmnMWAsLf60VgV83dft3ZQrmagWHMGDow6wJ-fboLw4NNmVe. Validated for WB, IP and cited by 52 publications. Rabbit anti-FIP200 (Cell Signaling Technology, Cat# 12436S): https://www.cellsignal.com/products/primary-antibodies/fip200-d10d11-rabbit-mab/12436?srsltid=AfmBOoqhJ2s2zL4bTCTli5bOsfPcOuEMBmKcTiFdB6phY4Ky3X9hwVpC. Validated for WB, IP and cited by 109 publications. HRP anti-Flag antibody (Abcam, Cat# b49763): https://www.abcam.com/en-us/products/primary-antibodies/hrp-ddddk-tag-binds-to-flag-tag-sequence-antibody-m2-ab49763?srsltid=AfmBOopiZ8XsS8G9AmaGtlndjKZ5NneaglYJrycjfEAYPdh98mIlB2vO. Validated for WB, ELISA and cited by over 85 publications. Donkey anti-goat IgG (Cy5) (Abcam, Cat# ab6566): https://www.abcam.com/en-us/products/secondary-antibodies/donkey-goat-igg-h-l-cy5-preadsorbed-ab6566?srsltid=AfmBOorvfsJG_O6TLDYUxlHzGeZq_SPdTLdTyoF263eElmnBi0barBTJ. Validated for WB and cited by 42 publications. Anti-GST Antibody (Cytiva, Washington DC, Cat# 27457701):https://www.cytivalifesciences.com/en/us/shop/chromatography/resins/affinity-tagged-protein/anti-gst-antibody-p-06000?srsltid=AfmBOoqIE2SS7jBtMby1gpFd0L0zbUIidO1toBETpcMuX73CQu2cpwMJ. Cited by 213 publications. Rabbit anti-MBP antibody (Thermo Fisher Scientific, Cat# PA1-989): https://www.thermofisher.com/antibody/product/Maltose-Binding-Protein-Antibody-Polyclonal/PA1-989. Validated for WB and cited by 9 publications. Alexa Fluor 488 goat anti-Riabbit IgG (Thermo Fisher Scientific, Cat# A11034): https://www.thermofisher.com/antibody/product/Goat-anti-Rabbit-IgG-H-L-Highly-Cross-Adsorbed-Secondary-Antibody-Polyclonal/A-11034. Validated for ICC/IF and cited by 7504 publications. Anti-HALO (Promega, Cat# G9211): https://www.promega.com/products/protein-detection/primary-and-secondary-antibodies/anti-halotag-monoclonal-antibody/?catNum=G9211. Validated for IF and cited by 114 publications. |

# Eukaryotic cell lines

Policy information about cell lines and Sex and Gender in Research

| | |
|---|---|
| Cell line source(s) | HEK293 GnTi cells were provided by the Cell Culture Facility of University of California, Berkeley. HeLa knockout cell lines generated in this study was submitted to Cellosaurus. |
| Authentication | All cell lines were validated by morphological analysis and routinely tested for absence of mycoplasma |
| Mycoplasma contamination | All cells were routinely tested for mycoplasma contamination using MycoAlert Mycoplasma Detection kit (Lonza, LT07-318). All cell lines were negative throughout the study. |
| Commonly misidentified lines (See ICLAC register) | No cell lines used in this study were found in the database of commonly misidentified cell lines that is maintained by ICLAC and NCB BioSample. |

## Plants

Seed stocks

N/A

Novel plant genotypes

N/A

Authentication

N/A

