## [Peer Review File · Nature Structural & Molecular Biology]

Structure and activation of the human autophagy-initiating ULK1C:PI3KC3-C1 supercomplex

Corresponding Author: Professor James Hurley

Version 0:

Decision Letter:

5th Nov 2024

Dear Dr. Hurley,

Thank you again for submitting your manuscript "Structure and activation of the human autophagy-initiating ULK1C:PI3KC3-C1 supercomplex". I apologize for the delay in this decision. I'm writing to let you know that we have decided to send your manuscript for peer review.

I am re-opening the manuscript submission link for you to resubmit your manuscript with all the associated files needed for the peer review process directly to our system, at your convenience. Please see below for details regarding the required materials. Please follow the link at the bottom of this email to upload the documents.

We want to ensure that the methods and statistics reporting in our papers are of the highest quality. To that end, we ask authors to fill out a Reporting Summary that collects information on experimental design and reagents, as well as an editorial Policy Checklist, which confirms compliance with our editorial policies, including the declaration of Competing Interests. If your paper includes ChIP-seq, flow cytometry or MRI data, we ask you take special care to complete those sections of the Reporting Summary as this data will aid greatly in the review of your manuscript.

These documents can be found by following the links below:

Reporting Summary:
<https://www.nature.com/documents/nr-reporting-summary.pdf>

Editorial Policy Checklist: <https://www.nature.com/documents/nr-editorial-policy-checklist.pdf>

Please be aware of our guidelines on digital image standards.

*****IMPORTANT*****

In order for us to proceed with the peer review process of your manuscript, we require you to provide accession numbers and reviewer tokens to access sequencing data sets. Please add this information to your manuscript file.

Please note we require official wwPDB validation reports for newly described atomic structures, as noted in the policy checklist. We also request that authors provide cryo-EM maps, half-maps and models, to help the reviewers in assessing the work. We recommend the use of figshare integration into our systems, which allows for provision of anonymous access links for the referees (<https://www.springernature.com/gp/authors/research-data/figshare-integration>).

Alternatively, please upload .zip folders directly with the submission. To ensure the ease of reviewer access to the data, please specify in the Data Availability section, where the files can be found (provide a figshare link or direct the reader to the manuscript files).

Additionally, I would like to kindly request that you provide the code used to analyse the data to the reviewers, if used. In order for the reviewers to evaluate the work adequately they must be able to test the software/review the code themselves. If you have not yet provided the software, we therefore request that you provide a single compressed zip file containing the software with a readme.txt file or other user manual containing complete instructions for installing and running the software. If appropriate, please also provide example data and expected output. Sufficient material should be provided for referees to

directly test the performance of the software/algorithm. If the software and materials are small enough to fit in a single compressed zip file less than 6MB in size, you may email this file directly to me. If the zip file is between 6 MB and 200 MB you may upload it to our file transfer site. If necessary, a second zip file up to 200 MB in size can be used to supply the example data. Please let me know if you need to use this option and I'll send you further details. Alternatively, you can also upload the code to GitHub and provide us with the link.

Please also fill out and return to me the code and software submission checklist that will be made available to editors and reviewers during manuscript assessment. Please note that this form is a dynamic 'smart pdf' and must therefore be downloaded and completed in Adobe Reader, instead of opening it in a web browser.

<https://www.nature.com/documents/nr-software-policy.pdf>

Please use the link below to submit the files. **Please also remember to move forward all other files associated with this version of the paper.**

Link Redacted

Sincerely,
Kat

Katarzyna Ciazynska, PhD
(she/her)
Senior Editor
Nature Structural & Molecular Biology
<https://orcid.org/0000-0002-9899-2428>

Version 1:

Decision Letter:

17th Dec 2024

Dear Dr. Hurley,

Thank you again for submitting your manuscript "Structure and activation of the human autophagy-initiating ULK1C:PI3KC3-C1 supercomplex". I apologize for the delay in responding, which resulted from the difficulty in obtaining suitable referee reports. Nevertheless, we now have comments (below) from the 3 reviewers who evaluated your paper. In light of those reports, we remain interested in your study and would like to see your response to the comments of the referees, in the form of a revised manuscript.

You will see that while reviewers appreciate the results, they raise several concerns which will need to be addressed in a revision. Specifically, reviewers #1 and #2 have concerns about the biological significance of the observed complexed and the reported stoichiometries. We ask that you attempt to further strengthen the conclusions, providing further functional validation, and confirm the interaction interfaces biochemically. While we agree that solidifying the cellular significance of the work will strengthen the manuscript, we appreciate that these experiments might prove to be out of scope of a revision, although the caveats of the experimental system used should be discussed.

Please be sure to address/respond to all concerns of the referees in full in a point-by-point response and highlight all changes in the revised manuscript text file. If you have comments that are intended for editors only, please include those in a separate cover letter.

[OPTIONAL: To guide the scope of the revisions, we list below a prioritized set of referee points that should be addressed in the revision, which we hope will be helpful to you. Please do not hesitate to contact me directly if you would like to discuss these issues further.

****Editor can outline here which reviewer concerns they are willing to overrule, and which must be addressed****

We expect to see your revised manuscript within 3-4 months. If you cannot send it within this time, please contact us to discuss an extension; we would still consider your revision, provided that no similar work has been accepted for publication at NSMB or published elsewhere.

Reporting Summary:

Please note that all key data shown in the main figures as cropped gels or blots should be presented in uncropped form, with molecular weight markers. These data can be aggregated into a single supplementary figure item. While these data can be displayed in a relatively informal style, they must refer back to the relevant figures. These data should be submitted with the final revision, as source data, prior to acceptance, but you may want to start putting it together at this point.

Data availability: this journal strongly supports public availability of data. All data used in accepted papers should be available via a public data repository, or alternatively, as Supplementary Information. If data can only be shared on request, please explain why in your Data Availability Statement, and also in the correspondence with your editor. Please note that for some data types, deposition in a public repository is mandatory - more information on our data deposition policies and available repositories can be found below:

<https://www.nature.com/nature-research/editorial-policies/reporting-standards#availability-of-data>

Nature Structural & Molecular Biology is committed to improving transparency in authorship. As part of our efforts in this direction, we are now requesting that all authors identified as 'corresponding author' on published papers create and link their Open Researcher and Contributor Identifier (ORCID) with their account on the Manuscript Tracking System (MTS), prior to acceptance. This applies to primary research papers only. ORCID helps the scientific community achieve unambiguous attribution of all scholarly contributions. You can create and link your ORCID from the home page of the MTS by clicking on

'Modify my Springer Nature account'. For more information please visit www.springernature.com/orcid.

Link Redacted

Sincerely,
Kat

Katarzyna Ciazynska, PhD
(she/her)
Senior Editor
Nature Structural & Molecular Biology
<https://orcid.org/0000-0002-9899-2428>

Referee expertise:

Referee #1: autophagy, structural biology

Referee #2: structural biology (cryo-EM)

Referee #3: autophagy, structural biology

Reviewers' Comments:

Reviewer #1 (Remarks to the Author):

Chen et al. present a structural analysis of the ULK1C complex, which is central to the initiation of autophagy. ULK1C is composed of the ULK1 kinase, the phosphoprotein ATG13, and the scaffold protein FIP200. Although previous studies, including those from the Hurley group, have visualized individual components of this complex, the precise architectural arrangement of these subunits within the intact ULK1C complex remained unresolved. In this work, the authors provide critical structural insights into how FIP200's N-terminus dimer engages one molecule of ULK1 and ATG13, and how this arrangement is integral to autophagy initiation.

By determining a cryo-EM structure, the authors reveal how the ULK1 MIT-ATG13 MIM unit interfaces directly with FIP200, forming the "2:1:1" complex stoichiometry. Their use of AlphaFold predictions, combined with HDX-MS experiments, elegantly demonstrates how the ATG13 intrinsically disordered region (IDR) binds FIP200, enabling the assembly of the functional ULK1C complex.

Furthermore, the authors capture and visualize by cryo-EM a transient "supercomplex" formed by ULK1C in association with the PI3KC3-C1 complex. They detail how the FIP200 N-terminus creates a composite interaction surface with VPS15, BECN1, and ATG14 of PI3KC3-C1, explaining the autophagy-specific nature of this interaction. Particularly noteworthy is the finding that BECN1 and ATG14 contribute to ULK1C binding, thus clarifying the roles of these autophagy-specific subunits of the PI3KC3-C1 complex.

An additional intriguing observation is that one of the ATG13 IDR regions bound to FIP200 modulates the ULK1C stoichiometry, shifting from the "2:1:1" to a "2:2:2" complex as concentration changes. In the authors' proposed model, enrichment of the "2:1:1" ULK1C complex at the autophagosome membrane by PI3KC3-C1 subsequently promotes ULK1 dimerization on FIP200, ultimately yielding the "2:2:2" state, where ULK1 is activated to initiate the autophagy cascade.

Overall, this study represents a substantial contribution to our understanding of the structural and mechanistic underpinnings of autophagy initiation. The authors have skillfully integrated various biochemical and structural approaches to navigate a complex, multifaceted problem. Their insights provide a significant advancement in the field and should be of broad interest. I commend the authors for this impressive work and recommend publication in Nature Structural & Molecular Biology.

Reviewer #2 (Remarks to the Author):

ULK1C and PI3KC3-C1 are essential for initiating human autophagy. In a study by Chen M. et al., the authors reported the cryo-EM structures of the ULK1C core and its complex with PI3KC3-C1. Combining cryo-EM with structural modeling and biochemical analysis, the study provides valuable insights into the autophagy initiation process at the molecular level. Overall, the manuscript is well-structured, presenting novel and significant results supported by robust experimental evidence.

Reviewer Comments:

Major Points:

While the study characterizes the molecular interactions between PI3KC3-C1 and ULK1C, it lacks biochemical validation of the interface. The authors are encouraged to conduct assays such as pulldown or GST-bead coating to further validate and investigate this interface.

Clarification is needed regarding the stoichiometry between PI3KC3-C1 and ULK1C. Is the binding of PI3KC3-C1 comparable to a 2:2:2 stoichiometry of ULK1C? Please provide corresponding structural or biochemical support.

What is the biochemical or cellular significance of the ULK1C 2:2:2 complex? The manuscript mentions that "Artificially induced dimerization of yeast Atg1 promotes its activation ... dimerization could be an autophagic cognate of the dimerization-based RTK activation paradigm." Could the authors test whether the 2:2:2 complex enhances kinase activity or if there is any cellular significance?

Minor Points:

In Fig. 1d, please change the dashed box label to lowercase for consistency.

In Fig. 1e, the authors should consider explaining the F466/F470 and L476/L479 lanes (controls?) in the figure legend.

In Fig. 2c:

- a) The pull-down domain boundary of ATG13 in lane 2 does not include the complete site #2. Since E403 is stated to be important but was not included in the pull-down experiment, please clarify.
- b) It would be beneficial to include pulldown data between FIP200NTD and ATG13(363-450 wt and 363-450 mut). Alternatively, is there a labeling confusion (similarly in Fig. 2f) as inferred from the figure legend?

Line 149 should reference Extended Data Fig. 5b, not Fig. 5b.

Reviewer #3 (Remarks to the Author):

The mammalian ULK complex and the class III phosphatidylinositol 3-kinase complex (PI3KC3-C1) are critically involved in autophagy initiation. How these complexes are working together is not well understood. Previously, the structural information of these two critical complexes are limited to low resolution EM maps. In this manuscript, Chen et al. report the cryo-EM structures of human ULK1 core complex, and an assembly involving the ULK1 core and PI3KC3-C1, at higher resolutions. These results reveal the molecular interactions organizing the ULK core complex although many are limited to the moderate quality of the cryo-EM map. Interestingly, they also uncover the physical interaction between the ULK core complex and PI3KC3-C1.

In spite of these merits, this study also shows several major limitations to the general readers of NSMB interested in the field, which should be addressed.

(1) As presented in the first paragraph of Results and Fig. 1a, large regions in each protein components of the ULK complex were deleted (or not included, ex. ATG101) to facilitate preparation and structural determination of the core complex. However, the structural information derived from this severely trimmed complex and its interaction with PI3KC3-C1 may contain artifacts and thus require to be validated by other means.

(2) Shown in Fig. 1c, is the FIP200-NTD dimer structurally symmetrical? If so, the authors need to discuss how the ATG13-MIM-UNK-MIT domain complex binds only to one monomer.

(3) Shown in Figs. 2a & b, if binding of the intrinsically disordered region (IDR) of ATG13 to the FIP200-NTD dimer would justify the formation of the 2:1:1 FIP200:ATG13:ULK1 complex, how could the three proteins form the 2:2:2 complex in full-length context?

(4) Shown in Fig. 7, there seems no evidence showing binding of the PI3KC3-C1 complex only to the 2:1:1 FIP200:ATG13:ULK1 complex. If this is indeed the case, there seems no evidence that the second ATG13/ULK1 molecules

may displace the bound PI3KC3-C1 complex from the 2:1:1 FIP200:ATG13:ULK1 complex.

(5) It is interesting to see how PI3KC3-C1 interacts with ULK1C1. But, as the resolution is limited, the authors should at least provide some biochemical evidence, ie. XL-MS or mutagenesis/pull down, to verify this assembly. For example, if the BECN3-BH3 and ATG14-BECN1-ZF are deleted, does it abolish the binding between FIP200 and PI3KC3-C1 and in vivo activity?

(6) No evidence were presented to support the existence of the 2:1:1 and/or the 2:2:2 FIP200:ATG13:ULK1 complexes in the cell.

(7) No evidence were presented showing that the 2:2:2 FIP200:ATG13:ULK1 complex is the functional assembly product as conveyed by Fig. 7.

Additional specific comments:

Fig. 1e: what are the corresponding hydrophobic residues on ULK1-MIT that interact with ATG13 MIM (L495, L499, F509, F512)? The residues should be labeled in Fig. 1e. Why did the authors introduce F466, F470, L476, L479 as negative control and where are these residues in the structure?

Figs. 2D,E,G,H are difficult to understand regarding the predicted locations of sites on ATG13-IDR in the ULK complex. Additionally, the IDR model should be presented in a more highlighting color.

Fig. 3: Please provide a quantification plot for the GST beads recruitment experiment.

ED Fig. 1d: The total number of particles used for 3D classification is around 2.2 millions, which is more than the number 1.9 millions from previous 2D class averages, please correct the number.

The clashscore value for PDB code: 9C82 seems too high. Model to map FSC for the reported structures in this study should be provided.

Fig. 4c: what does the label "G" pointing to FIP200 mean?

Line 953: 20 degree.

ED Fig. 6e, ab-initio should be ab-initio.

Line 163, please indicate residue 166/181 and 473/485 of FIP200 in the corresponding figure.

In the materials and methods section, the terminology is not consistent in the text. In the text, "site M" is used while in line 565, "site 5" is used. Additionally, paragraphs including "Generation of knockout lines using CRISPR/Cas9" and "Cloning and generation of stable cell lines" etc. described some experiments that were not presented in this study. The author should carefully go through and revise.

Version 2:

Decision Letter:

Our ref: NSMB-A49984B

10th Feb 2025

Dear Dr. Hurley,

Thank you for submitting your revised manuscript "Structure and activation of the human autophagy-initiating ULK1C:PI3KC3-C1 supercomplex" (NSMB-A49984B). It has now been seen by the original referees and their comments are below. I am writing to let you know that in the light of their comments, we'll be happy in principle to publish it in Nature Structural & Molecular Biology, pending minor revisions to satisfy the referees' final requests and to comply with our editorial and formatting guidelines.

We are now performing detailed checks on your paper and will send you a checklist detailing our editorial and formatting requirements in about 2-3 weeks. Please do not upload the final materials and make any revisions until you receive this additional information from us.

Sincerely,
Kat

Katarzyna Ciazynska, PhD
(she/her)
Senior Editor
Nature Structural & Molecular Biology
<https://orcid.org/0000-0002-9899-2428>

Reviewer #1 (Remarks to the Author):

As I wrote for the first round of the review, I support the publication of this work in NSMB. I feel the significance and quality of the work is at least the same level of the many papers NSMB has published, justifying this recommendation.

Reviewer #2 (Remarks to the Author):

As a reviewer, I appreciate the authors' thoughtful response, and the significant efforts made to address the reviewers' concerns. The challenges outlined regarding the limitations of complementary techniques in keeping pace with cryo-EM advancements are well-articulated and reflect a broader issue in the field indeed. I agree that delaying progress while awaiting new methods for functional in-cell confirmation is neither practical nor productive. Their transparency in distinguishing data-driven suggestions from firm conclusions ensures the manuscript remains scientifically rigorous and appropriately cautious. While some proposed experiments were unfeasible, the authors have clearly explained these limitations and incorporated necessary caveats.

Overall, I find the authors' response thorough and reasonable. The manuscript, with its ample and sophisticated data, represents a significant contribution to the field, offering far-reaching implications that will inspire future studies. I support its publication in its current form.

Reviewer #3 (Remarks to the Author):

In this revised manuscript, almost all of my major concerns regarding this work are not addressed. Therefore, this work present more questions than providing answers to the field.

Version 3:

Decision Letter:

11th Apr 2025

Dear Dr. Hurley,

We are now happy to accept your revised paper "Structure and activation of the human autophagy-initiating ULK1C:PI3KC3-C1 supercomplex" for publication as an Article in Nature Structural & Molecular Biology.

Your paper will be published online soon after we receive proof corrections and will appear in print in the next available issue. You can find out your date of online publication by contacting the production team shortly after sending your proof corrections.

Sincerely,

Katarzyna Ciazynska, PhD
(she/her)
Senior Editor
Nature Structural & Molecular Biology
<https://orcid.org/0000-0002-9899-2428>

We thank all of the reviewers for their time and insights. We apologize for being unable to carry out all of the proposed experiments. There is core issue that is not unique to our situation, but is increasingly common in cryo-EM. The power of single particle cryo-EM to provide insights into conformational ensembles and rare and transient states has outrun the ability of complementary techniques to confirm them. Further advances in robust and accessible single molecule or near-single molecule imaging in cells will be needed, we believe, for functional in-cell confirmation to keep up with progress in cryo-EM. It would be wrong to hold the field back pending the development of such methods.

The first version of this manuscript was posted to bioRxiv in June, 2023. From then through Oct., 2024, the manuscript was under review and revision at *Science*, and many additional data were added in the course of that process, which were carried over to this version. Ultimately, several experiments suggested by the reviewers at *Science* proved to be unfeasible. Several of these experiments were also suggested by the reviewers of this version of the manuscript. All of these had already been attempted, with major investments of effort. We have been able to respond quickly to the present round of review since such extensive efforts had already been made over the course of the past year to address the points concerning functional confirmation of the stoichiometry change in cells and related issues. This paper already contains ample and sophisticated corroborating data using rescue of *ATG13* CRISPR KO cells and cryo-EM and mass photometry studies of variant complexes. In a study of multiple new structures with far-reaching implications, it should not be necessary to investigate and corroborate every implication of every structure. Surely, some points can reasonably be left for future studies. Details concerning specific suggestions and experimental outcomes are provided below.

We hope the editor and reviewers will understand we have tried our best to carry out every one of the proposed experiments. Where this was not possible, in each case the appropriate caveat has been incorporated into the text. Here, we followed the editor's guidance that "While we agree that solidifying the cellular significance of the work will strengthen the manuscript, we appreciate that these experiments might prove to be out of scope of a revision, although the caveats of the experimental system used should be discussed." In the current revision, concepts suggested (as opposed to proven) by the data are identified as suggestions rather than conclusions. No firm conclusions are drawn that are not fully supported by corroborating data.

Reviewers' Comments:

Reviewer #1 (Remarks to the Author):

Chen et al. present a structural analysis of the ULK1C complex, which is central to the initiation of autophagy. ULK1C is composed of the ULK1 kinase, the phosphoprotein ATG13, and the scaffold protein FIP200. Although previous studies, including those from the Hurley group, have visualized individual components of this complex, the precise architectural arrangement of these subunits within the intact ULK1C complex

remained unresolved. In this work, the authors provide critical structural insights into how FIP200's N-terminus dimer engages one molecule of ULK1 and ATG13, and how this arrangement is integral to autophagy initiation.

By determining a cryo-EM structure, the authors reveal how the ULK1 MIT-ATG13 MIM unit interfaces directly with FIP200, forming the "2:1:1" complex stoichiometry. Their use of AlphaFold predictions, combined with HDX-MS experiments, elegantly demonstrates how the ATG13 intrinsically disordered region (IDR) binds FIP200, enabling the assembly of the functional ULK1C complex.

Furthermore, the authors capture and visualize by cryo-EM a transient "supercomplex" formed by ULK1C in association with the PI3KC3-C1 complex. They detail how the FIP200 N-terminus creates a composite interaction surface with VPS15, BECN1, and ATG14 of PI3KC3-C1, explaining the autophagy-specific nature of this interaction. Particularly noteworthy is the finding that BECN1 and ATG14 contribute to ULK1C binding, thus clarifying the roles of these autophagy-specific subunits of the PI3KC3-C1 complex.

An additional intriguing observation is that one of the ATG13 IDR regions bound to FIP200 modulates the ULK1C stoichiometry, shifting from the "2:1:1" to a "2:2:2" complex as concentration changes. In the authors' proposed model, enrichment of the "2:1:1" ULK1C complex at the autophagosome membrane by PI3KC3-C1 subsequently promotes ULK1 dimerization on FIP200, ultimately yielding the "2:2:2" state, where ULK1 is activated to initiate the autophagy cascade.

Overall, this study represents a substantial contribution to our understanding of the structural and mechanistic underpinnings of autophagy initiation. The authors have skillfully integrated various biochemical and structural approaches to navigate a complex, multifaceted problem. Their insights provide a significant advancement in the field and should be of broad interest. I commend the authors for this impressive work and recommend publication in Nature Structural & Molecular Biology.

We are most grateful for the reviewer's thorough analysis and appreciation of the work.

Reviewer #2 (Remarks to the Author):

ULK1C and PI3KC3-C1 are essential for initiating human autophagy. In a study by Chen M. et al., the authors reported the cryo-EM structures of the ULK1C core and its complex with PI3KC3-C1. Combining cryo-EM with structural modeling and biochemical analysis, the study provides valuable insights into the autophagy initiation process at the molecular level. Overall, the manuscript is well-structured, presenting novel and

significant results supported by robust experimental evidence.

Thank you very much for this positive evaluation.

Reviewer Comments:

Major Points:

While the study characterizes the molecular interactions between PI3KC3-C1 and ULK1C, it lacks biochemical validation of the interface. The authors are encouraged to conduct assays such as pulldown or GST-bead coating to further validate and investigate this interface.

We agree this would be desirable. This has been challenging because the moderate affinity interaction is distributed over a large surface area. FIP200 residues in the interface with the VPS15 solenoid include Ala39, Ile40, Leu80, Phe457, and Leu460. VPS15 residues include Val566, Ala606, Leu645, Leu647, and Ile686. Additional interactions involve FIP200 and the N-terminal zinc binding regions of BECN1 and ATG14, which are more mobile and therefore the pairwise interactions cannot be modeled atomistically with high confidence. Double hydrophobic to Asp mutants of FIP200 and a triple hydrophobic to Asp mutant of VPS15 were generated to disrupt the interface. FIP200 A39D/I40D and LL457-460DD, and VPS15 A606D/LL645-647DD were generated and bead binding assays were carried out. Binding was seen at essentially wild-type levels. In order to disrupt the interface more severely, a quadruple mutant of FIP200 and quadruple and quintuple mutants of VPS15 were generated, however, none of these generated soluble protein. Thus, the distributed nature of the binding site over a large area requires a large number of mutations to disrupt it. Yet the marginally stable nature of these proteins, which are naturally expressed in cells at very low levels and already require special care in their biochemical handling, leads to a loss in stability upon extensive mutagenesis.

We have added the caveat at line 225 that “We note that the ULK1C:PI3KC3-C1 interface is distributed over a large region of the surfaces of these complexes, which has made it challenging to engineer stable mutants that selectively block the interaction.”

Clarification is needed regarding the stoichiometry between PI3KC3-C1 and ULK1C. Is the binding of PI3KC3-C1 comparable to a 2:2:2 stoichiometry of ULK1C? Please provide corresponding structural or biochemical support.

Our study focuses on a 1:2:1:1 PI3KC3-C1:FIP200:ATG13:ULK1 complex, which is what is observed as a substantial population by cryo-EM. With respect to whether a 1:2:2:2 PI3KC3-C1:FIP200:ATG13:ULK1 complex exists, this would be very clear in the cryo-EM if it existed, but we have seen no evidence for even a minor population

What is the biochemical or cellular significance of the ULK1C 2:2:2 complex? The manuscript mentions that "Artificially induced dimerization of yeast Atg1 promotes its activation ... dimerization could be an autophagic cognate of the dimerization-based RTK activation paradigm." Could the authors test whether the 2:2:2 complex enhances kinase activity or if there is any cellular significance?

We endeavored to test this important question by incubating purified ULK1C with PI3KC3-C1 or the dimer-promoting ATG13 construct and testing ULK1 autophosphorylation with anti-pT180. Unfortunately, purified ULK1 within ULK1C is already phosphorylated at this site. Dephosphorylation of recombinant ULK1C with lambda phosphatase resulted in inactive material that could not be rephosphorylated, even with exogenous active ULK1. We attempted this experiment with ULK1 and dimer-promoting truncated ATG13 in cells, however, the available antibody provides a weak signal that is not suitable for analysis of cell lysates, only for highly concentrated recombinant material. Therefore, the answer to this interesting question remains for now out of reach.

Minor Points:

In Fig. 1d, please change the dashed box label to lowercase for consistency.

Corrected

In Fig. 1e, the authors should consider explaining the F466/F470 and L476/L479 lanes (controls?) in the figure legend.

We added panels in Extended Data Fig. 2 with the corresponding description at line 103 that "Four pairs of hydrophobic residues (F466/F470, L476/L479, L495/L499, and F509/F512) from the ATG13^{MIM} form prominent direct interactions with ULK1^{MIT} (Extended Data Fig. 2d and 2e). Mutagenesis and pull-down assays confirmed that the latter two pairs, L495/L499, and F509/F512, are critical for the binding (Fig. 1e)."

In Fig. 2c:

a) The pull-down domain boundary of ATG13 in lane 2 does not include the complete site #2. Since E403 is stated to be important but was not included in the pull-down experiment, please clarify.

Fig. 2c is designed to show the progressive mapping of the binding determinants in site 2. The construct containing part of site 2 through residue 398 is sufficient for binding as shown in lane 2. In lane 3, deletion of essentially all of site 2 in the construct ending at 392 eliminates binding, showing site 1 alone is insufficient for binding, and that in the context of site 2, residues 393-398 are necessary for binding. The purpose of the experiment shown in lane 4 is to determine what mutation within site 2, in the context of a longer portion of the IDR, is minimally sufficient to prevent binding. This is a

necessary step towards determining the mutation needed to knock out site 2 in the cell assay shown in panels j and l. In this longer construct, we found that it is necessary to combine E403K with mutation of residues 394, 396 and 397 in the core of site 2 in order to completely eliminate binding. Full details of the intermediate experiments are not reported as they were done on a screening basis and have no bearing on the interpretation.

Overall, experiments in lanes 1-3 are intended to map the core of site 2. On the other hand, the experiments in lanes 4-5 have separate purposes, which are to validate the structural observations and to set the stage for disruption of ATG13 site 2 in the cell-based assays.

In the interest of clarity, we re-aligned the bars above the gel for wt and mutant FIP200. We also replaced the ATG13 label “Mut” with a more specific designation of “M1”, and indicate the residues comprising M1 below the panel for the reader’s convenience.

b) It would be beneficial to include pulldown data between FIP200NTD and ATG13(363-450 wt and 363-450 mut). Alternatively, is there a labeling confusion (similarly in Fig. 2f) as inferred from the figure legend?

The logic of Fig. 2f follows a similar flow to that of 2c. Lane 1 shows that the most C-terminal portion of site 2, and sites 3, and 4 are sufficient for binding. Lane 2 shows that the C-terminal part of site 2 plus site 3 is insufficient. Lane 3 shows that site 4 is sufficient, ergo site 3 is not necessary when sites 4 is present. Having mapped the minimal determinant for sufficiency in the C-terminal segment to site 4, the remaining two lanes validate the structurally implied roles of residues FIP200 V44, and ATG13 A452 and F453. As in Fig. 2c, a variety of single and smaller subsets that were screened and found insufficient to knock out the interaction are not shown because they were not judged to be of broad interest nor to impact the conclusions. As in Fig. 2c, we realigned the bars and designated the mutant as “M2” instead of “Mut” and defined M2 below the rest of the panel.

Line 149 should reference Extended Data Fig. 5b, not Fig. 5b.

Corrected

Reviewer #3 (Remarks to the Author):

The mammalian ULK complex and the class III phosphatidylinositol 3-kinase complex (PI3KC3-C1) are critically involved in autophagy initiation. How these complexes are working together is not well understood. Previously, the structural information of these two critical complexes are limited to low resolution EM maps. In this manuscript, Chen et al. report the cryo-EM structures of human ULK1 core complex, and an assembly involving the ULK1 core and PI3KC3-C1, at higher resolutions. These results reveal the

molecular interactions organizing the ULK core complex although many are limited to the moderate quality of the cryo-EM map. Interestingly, they also uncover the physical interaction between the ULK core complex and PI3KC3-C1.

In spite of these merits, this study also shows several major limitations to the general readers of NSMB interested in the field, which should be addressed.

(1) As presented in the first paragraph of Results and Fig. 1a, large regions in each protein components of the ULK complex were deleted (or not included, ex. ATG101) to facilitate preparation and structural determination of the core complex. However, the structural information derived from this severely trimmed complex and its interaction with PI3KC3-C1 may contain artifacts and thus require to be validated by other means.

We previously characterized full-length and near full-length versions of the ULK1 complex in Shi et al., J. Cell Biol., 2020 by negative stain EM and HDX-MS. The regions mentioned are IDRs or linked to the core by IDRs and were found by NS-EM not to be ordered relative to the core.

With respect to the possibility that PI3KC3-C1 interacts with regions of ULK1C outside of the trimmed core complex, we did test an interaction with the ATG13-ATG101 HORMA dimer that had been previously reported to bind to a subcomplex of PI3KC3-C1 by Nguyen et al. (2023), and found that it did not bind to the full PI3KC3-C1, data which are included in the manuscript. At a minimum, it is already known that the kinase domain of ULK1 must interact, since all four subunits of PI3KC3-C1 are known substrates of ULK1 phosphorylation. We cannot exclude that there may be some additional interactions between PI3KC3-C1 and yet other portions of ULK1C under some conditions, although there is no specific reason to expect them. Given the absence of any particular evidence or rationale, a systematic search for such interactions would in a sense be a "fishing trip" and beyond the scope of this study.

(2) Shown in Fig. 1c, is the FIP200-NTD dimer structurally symmetrical? If so, the authors need to discuss how the ATG13-MIM-UNK-MIT domain complex binds only to one monomer.

The two protomers of FIP200^{NTD} in Fig. 1c are not structurally symmetrical. The one that binds to ULK1^{MIT}:ATG13^{MIM} adopts a more compact conformation than the other (see Extended Data Fig.1), which enabled us to achieve a higher resolution comparing to the previous FIP200^{NTD} dimer alone structure (Shi et al., J. Cell Biol., 2020). Moreover, regardless of whether the ULK1^{MIT}:ATG13^{MIM} subunits bind to the left or right FIP200^{NTD} protomer, the protein assembly was treated identically during cryoEM processing. All the FIP200^{NTD} protomers bound to the ULK1^{MIT}:ATG13^{MIM} were aligned to the same side, resulting in the 2:1:1 structure presented in this study. The 2:2:2 complex was not formed, as the second ULK1^{MIT}:ATG13^{MIM} was not observed during the 3D classification and subsequent steps.

(3) Shown in Figs. 2a & b, if binding of the intrinsically disordered region (IDR) of ATG13 to the FIP200-NTD dimer would justify the formation of the 2:1:1 FIP200:ATG13:ULK1 complex, how could the three proteins form the 2:2:2 complex in full-length context?

This is described in full detail in the section on “**FIP200 scaffolding of ULK1 dimerization**”.

(4) Shown in Fig. 7, there seems no evidence showing binding of the PI3KC3-C1 complex only to the 2:1:1 FIP200:ATG13:ULK1 complex. If this is indeed the case, there seems no evidence that the second ATG13/ULK1 molecules may displace the bound PI3KC3-C1 complex from the 2:1:1 FIP200:ATG13:ULK1 complex.

If a 1:2:2:2 PI3KC3-C1:FIP200:ATG13:ULK1 complex existed, this would have been very clear in the cryoEM map. Please refer to our response to review 2 above.

The model we propose suggests that PI3KC3-C1 displaces the ATG13/ULK1-dimer-inhibiting ATG13 site 2. We agree with the reviewer's inference that the model implies that a second ATG13/ULK1 should displace bound PI3KC3-C1 in order to liberate it and generate the PI3KC3-C1-free 2:2:2 complex, but we do not actually claim this in the manuscript. It does not seem necessary to provide supporting evidence in this case, since we have been careful to avoid making this claim.

(5) It is interesting to see how PI3KC3-C1 interacts with ULK1C1. But, as the resolution is limited, the authors should at least provide some biochemical evidence, ie. XL-MS or mutagenesis/pull down, to verify this assembly. For example, if the BECN3-BH3 and ATG14-BECN1-ZF are deleted, does it abolish the binding between FIP200 and PI3KC3-C1 and in vivo activity?

We have made the necessary constructs and found that deleting the ATG14 and BECN1 ZNF regions led to a loss of PI3KC3-C1 protein expression in our hands. We added at line 248 the statement “As a limitation, we note that it was not possible to test complex formation by N-terminally truncated forms of ATG14 and BECN1 due to loss of stability. Moreover, the local resolution of the interface in the region involving ATG14 and BECN1^{BH3} was insufficient to atomistically model molecular contacts.”

(6) No evidence were presented to support the existence of the 2:1:1 and/or the 2:2:2 FIP200:ATG13:ULK1 complexes in the cell.

This is an exceptionally difficult task since the complex is expected to exist transiently at autophagy initiation sites that contain only a few tens of molecules and exist transiently for, most likely, just a few minutes, all in the context of a much larger number of diffuse soluble molecules. To illustrate the challenge, see the papers by Broadbent et al, J Cell Biol. (2023), and Banerjee, Sci Adv. (2023). We believe a full answer would require

doing a three color version (monitoring ATG14, FIP200, and ULK1) using ATG13 mutants expressed at endogenous levels in cells with single molecule precision in quantitation. Ultimately we hope that an improved to the ATG13 mutants could provide the needed evidence, however, this will be a substantial undertaking beyond the scope of the current manuscript. At line 267 we now state “The small number of molecules involved and the transient nature of autophagy initiation has thus far precluded direct imaging of ULK1 dimerization, however.”

(7) No evidence were presented showing that the 2:2:2 FIP200:ATG13:ULK1 complex is the functional assembly product as conveyed by Fig. 7.

Please see the response to point 6 above.

Additional specific comments:

Fig. 1e: what are the corresponding hydrophobic residues on ULK1-MIT that interact with ATG13 MIM (L495, L499, F509, F512)? The residues should be labeled in Fig. 1e. Why did the authors introduce F466, F470, L476, L479 as negative control and where are these residues in the structure?

Two panels were added to the Extended Data Fig 2 (d and e) to describe the interface between ULK1^{MIT} and ATG13^{MIM}.

To clarify the choice of F466/F470 and L476/L479, we added a description at line 103 that “Four pairs of hydrophobic residues (F466/F470, L476/L479, L495/L499, and F509/F512) from the ATG13^{MIM} form prominent direct interactions with ULK1^{MIT} (Extended Data Fig. 2d and 2e). Mutagenesis and pull-down assays confirmed that the latter two pairs, L495/L499, and F509/F512, are critical for the binding (Fig. 1e).”

Figs. 2D,E,G,H are difficult to understood regarding the predicted locations of sites on ATG13-IDR in the ULK complex. Additionally, the IDR model should be presented in a more highlighting color.

The predicted locations of Fig. 2d, e, g, h are indicated in the overview Fig. 2a using dashed boxes. The IDR model is highlighted in yellow, in contrast to FIP200, which is colored in blue.

Fig. 3: Please provide a quantification plot for the GST beads recruitment experiment.

The GST beads assay is designed to qualitatively characterize the binding properties of each ULK1C construct against PI3KC3-C1. We focus only on whether binding occurs here, and measuring the binding strength is not intended in this study.

ED Fig. 1d: The total number of particles used for 3D classification is around 2.2

millions, which is more than the number 1.9 millions from previous 2D class averages, please correct the number.

Corrected

The clashscore value for PDB code: 9C82 seems too high. Model to map FSC for the reported structures in this study should be provided.

We adopted a poly-Ala model to mitigate the clash, however, the score did not further improve due to the limitations of the EM map, i.e., the moderate resolution (8.03 Å) and the preferred orientation problem (see Extended Data Fig.6). The model to map FSC was calculated by the Cryo-EM Validation tools of Phenix-1.21.1, as shown below, indicating that the map interpretation is adequate.

Fig. 4c: what does the label "G" pointing to FIP200 mean?
The label was removed.

Line 953: 20 degree.
Corrected.

ED Fig. 6e, ab-initio should be ab-initio.
Corrected.

Line 163, please indicate residue 166/181 and 473/485 of FIP200 in the corresponding figure.

A panel was added to Extended Data Fig.7 to indicate the interface. We also named the two interfaces as interface 1 and 2 at line 167 that “The interface 1 is formed between the FIP200^(166-181/473-485) and BECN1^{BH3} (Extended Data Fig. 7b), ... The interface 2 is formed by the globular ULD domain of FIP200 and the curved HSD domain of VPS15 (Extended Data Fig. 7c).”

In the materials and methods section, the terminology is not consistent in the text. In the text, "site M" is used while in line 565, "site 5" is used. Additionally, paragraphs including "Generation of knockout lines using CRISPR/Cas9 " and "Cloning and generation of stable cell lines" etc. described some experiments that were not presented in this study. The author should carefully go through and revise.

The terminology was corrected and the methods were updated to address these comments.